# Robust determinants of income distribution across and within countries

**Liang Frank Shao** [ORCID] *

School of Economics, Henan University, Kaifeng, China

* Mingliangshao@foxmail.com

## Abstract

Multicollinearity widely exists in empirical studies, which leads to imprecise estimation and even endogeneity when omitted variables are correlated with any regressors. We apply an innovative strategy, different from the usual tools (instrumental variable, ridge regression, and least absolute shrinkage and selection operator), to estimate the robust determinants of income distribution. We transform panel data into (quasi-) cross-sectional data by removing country and time effects from the data so that all variables become zero mean and orthogonal to the country dummies and time variable, and multicollinearity becomes very low or even disappears with the quasi-cross sectional data in any specifications regardless of country dummies and time variable being included or not. Our contribution is threefold. First, we build a general method to address the multicollinearity issue in panel data, which is to isolate the common contents of correlated variables and ensures robust estimates in different specifications (dynamic or static specifications) and estimators (within- or between-effects estimators). Second, we find no evidence for the Kuznets hypothesis within and across countries; investment is economically and statistically the most robust determinant of income inequality; meanwhile, labor income share shows robustly and consistently positive effects on income inequality, which challenges the related literature. Last, simulations with our estimates show that the total marginal effects of development (regarding GDP, capital stock and investment) on income inequality are very likely to be positive within and between countries except that the impacts on middle-60% and top-quintile income shares are not so likely to increase income inequality across countries.

## I Introduction

Multicollinearity often exists among the variables with panel data and those in structured function forms, and it mainly results in two issues in empirical studies. One is imprecision of estimation on correlated variables; the other is endogeneity when omitted variables are relevant and correlated with those concerned. Instrumental variables are often used to address endogeneity issues, and the validity of an instrument relies on its correlation strength with the instrumented variables and its noncorrelation strength with the error terms, which may not be perfectly confirmed when using panel data because the estimated residuals are also panel data that are often highly correlated with covariates. The other strategy is either to shrink large

**Data Availability Statement:** All relevant data are within the manuscript and its Supporting Information files.

**Funding:** The author(s) received no specific funding for this work.

**Competing interests:** The authors have declared that no competing interests exist.

coefficients (ridge regression) or select the most related covariates (least absolute shrinkage and selection operator, LASSO), which are also not good solutions because either multicollinearity is not properly removed or relevant regressors are not considered in the regressions.

Panel data has been intensively and directly used in the empirical studies [1–4] of income distribution. But multicollinearity has often been ignored in the literature so that omitted variable bias and endogeneity have been the main obstacles that are hardly well treated, and robust determinants of income distribution are still an open question in the literature.

In this paper, we transfer panel data into quasi-cross-sectional data by extracting country-fixed effects and time (fixed or trended) effects from panel data, which is similar to removing the transportation channels (country-fixed effects and time effects) of multicorrelation among explanatory variables. As a result, all variables that include GDP terms are almost, if not perfectly, isolated from each other, and multicollinearity is dramatically reduced or even completely removed. With the quasi-cross-sectional data, the risks of missing variable bias and the endogeneity caused by omitted covariates are effectively reduced or even perfectly removed.

The rationality of this method [3, 5] is that each variable with panel data has been partitioned into two orthogonal parts, one is for the variable itself, the other is for the time and section effects, which still reserve the full information of the panel data when both parts are considered in a regression. The advantage is that we still consider all covariates, among which multicollinearity has been effectively reduced so that estimates become more robust and precise comparing the existing tools.

We take two steps to obtain the quasi-cross-sectional data. First, we decompose GDP into two orthogonal proportions: explained GDP and unexplained GDP. Explained GDP is the fitted value from the ordinary least squares (OLS) regression of GDP on all known variables, including inequality variables, country dummies, and time variables. The OLS regression residuals are defined as unexplained GDP, which is orthogonal to the predicted (explained) GDP and all explanatory variables, and it will be used to explain income distribution as well. Second, we run a simple OLS regression on country dummies and time (dummy or trend) for each variable of the panel data and save the residuals for the regressions, which form the quasi-cross-sectional data for the regressed variables.

All variables including explained GDP in the quasi-cross-sectional data set have zero mean and are orthogonal to unexplained GDP, and bilateral correlations between any explanatory variables become very low or even near zero. We consider both fixed and trend effects for the time variable, and we run static and dynamic specifications and end up with choosing the one that gives the most robust and consistent results in the estimations.

We consider seven statistical measures of income distribution: the Gini coefficient (Gini), the bottom (B20) and top (T20) quintiles of income share, the middle 60% income share (MID), the income share below the median income (MES), the income share below the mean income (MIS), and the mean population share (MPS), that is, for the individuals whose income is below the mean income. MPS and MIS are used in Shao and Krause [6] to discuss if rising mean incomes had favored the middle income earners in relative terms.

We consider the time variable in data transfer and the estimations as either fixed or trended and use a simple least squares dummy variable (LSDV) and pooled OLS (POLS) regressions for the six measurements on the quasi-cross-sectional data.

We find three sets of variables that robustly and consistently impact income inequality differently. The first includes variables that negatively impact inequality (employment, primary education, etc.); the second includes variables that positively enhance inequality (investment, labor income share, etc.); and the third includes civil liberty and openness, whose roles rely on the development level of a country.

Our contribution is threefold. First, we build a general method to address the endogeneity issues caused by omitted variables being correlated with covariates when using panel data, which ensures robust results in different specifications and estimators (dynamic or static specifications and within- or between-effects estimators). Second, we update the literature on robust determinants of income distribution within and/or across countries; no evidence is found for the Kuznets hypothesis, labor income share (openness) is positively (negatively) associated with income inequality, and investment, tertiary education and the unexplained GDP are the three most robust determinants of increasing income inequality, and the most robust determinants of decreasing income inequality are employment, explained GDP, working hours, interactive term of openness with GDP, and primary education. Last, simulations with our estimates show that the total marginal effects of GDP, capital stock and investment on income inequality are very likely to be positive.

The paper proceeds as follows. Section II briefly reviews the literature. Section III discusses the data and shows how bilateral correlation can be effectively reduced by transferring panel data into quasi-cross-sectional data. Section IV presents a replication of three typical studies. Section V provides our unbiased, consistent and robust results, and Section VI concludes the paper.

## II Literature review

### 2.1 The Kuznets hypothesis

Since Kuznets [7] proposed the famous hypothesis that income inequality shows an inverted U-shape in GDP per capita, this hypothesis has been debated in theoretical and empirical studies [8–12].

Some studies [1, 13–17] find strong evidence for the hypothesis using panel data. Huang [18] presents evidence for the hypothesis using a reduced function form and a cross-country dataset. These estimations suffer serious endogeneity issues caused by omitting correlated-covariates and imprecise estimates.

There is also empirical evidence [2, 19] challenging the hypothesis. Savvides and Stengos [20] support the hypothesis with the quadratic form of GDP using threshold regression and pooled OLS (POLS) estimation on panel data, but the results are not robust when country dummies are considered. Frazer [21] finds various inequality-development relationships depending on the choice of years and countries, but country- and time-fixed effects are not included in his estimation.

Kalliovirta and Malinen [22] show that USA inequality drives inequality in other developed countries; which shows evidence for the structural correlation of panel data and the time correlation among all variables in the panel data. Therefore, all of the above studies based on panel data face serious endogeneity issues caused by missing correlated covariates.

### 2.2 Openness

Openness to international trade has been argued to be a major determinant of income inequality [17, 23, 24], but the empirical evidence is mixed. Some authors report no significant effect of openness on inequality [2, 25], while others find a positive effect, which is stronger for poorer countries [1, 24, 26, 27]. A negative correlation between openness and the labor income share is found in Harrison (2005) [28] and Ortega and Rodriguez (2006) [29]; Higgins and Williamson (2002) [15] find only limited support for openness increasing inequality.

Milanovic (2005) [24] uses the data of Deininger and Squire (1997) [30] and finds that the interactive effect of openness and mean income is positive on the income shares of the poor and the middle class but negative on the income share of the rich in his estimations. However, Milanovic's estimations also suffer serious missing-variable bias because openness is highly

correlated with capital stock, investment, financial development, etc., which are not considered in his regressions.

Jaumotte et al. (2013) [31] show that technological progress has more impacts on rising income inequality than globalization does, and the limited overall impacts of globalization are reflected by two offsetting forces: negative effects of trade and positive effects of financial globalization on inequality. We let explained GDP and unexplained GDP denote technological progress. Openness and financial development are also considered in our estimation. The largest bilateral correlation among all these variables in quasi-cross-sectional data is only -0.177, which is between openness and unexplained GDP.

It has also been noted [32, 33] that openness is negatively related to labor income share using the latest 5-year average panel data, which is in line with our finding.

## 2.3 Education

Different levels of educational attainment, primary, secondary and tertiary, may have various effects on income distribution; empirical studies using single average educational attainment (Thomas et al., 2000; Checchi and Garcı´a-Pen˜alosa, 2010) [34, 35] cannot reveal the difference between these effects. Castello-Climent and Domenech (2017) [36] find that the wage Gini has an inverted U-shape in terms of the human capital Gini, which might imply inconsistent effects of various levels of educational attainment on income inequality.

Li et al. (1998) [2] point out that initial secondary schooling is an important determinant of inequality, which is not supported by our estimations; we find that all three levels of educational attainment do have robust and significant effects on income distribution, which are also reported in Barro (2000) [1].

Eicher and Garcia-Penalosa (2001) [37] analyze the non-monotonic relation between educational attainment and inequality. Recent studies (Erosa, et al., 2010; Santos, Sequeira and Ferreira-Lopes, 2017) [38, 39] show that income inequality is significantly affected by human capital accumulation and TFP. We use three levels of educational attainment to denote human capital because tertiary education could play different roles than primary and secondary educations in inequality. We do not take TFP as an explanatory variable to avoid its high correlation with productive factors.

## 2.4 Labor income share

Labor income share is correlated with many factors, for instance [40, 41]. Many studies (Daudey and García-Peñalosa, 2007; Checchi and García-Peñalosa, 2010; Bengtsson and Waldenström, 2018) [35, 42, 43] have documented the strongly positive (negative) correlation between capital (labor) income share and top personal income shares (the Gini coefficient) and global decline of labor share has also been noted [44]; which contradict our results that labor income share is positively associated with income inequality within and across countries.

The effect of capital income share on income distribution is explained by the effects of capital stock and investment in our estimation. The effect of capital stock is negative and economically small, and the effect of investment is positive and economically large on income inequality. Any changes in labor income share might be dominated by changes in the earnings of top income earners because labor income share is positively (negatively) correlated with the top (bottom) 20% income share in our regressions, and the bottom income shares (B20, MED, MIS) are relatively stable in the data.

## 2.5 Other factors

Some studies (Schultz, 1998; Li, Squire and Zou, 1998; Barro, 2000) [1, 2, 45] demonstrate that income inequality is explained mainly by country variations and very little by time variation,

which implies that the dynamic LSDV specification might be an appropriate choice to explain within-country variations, where the country variations are explained by the country dummies and the lagged inequality.

We consider many other factors discussed in the literature, which include civil liberty, initial secondary schooling, financial depth, initial land inequality (Li, Squire and Zou, 1998) [2], democracy indices (Rodrik, 1999) [46–48], economic freedom (Carter, 2006) [49], population growth (Deaton and Paxson, 1997) [50], inflation (Bulir, 2001) [51], etc. and unemployment (Jantti and Jenkins, 2010) [52] Capital stock and investment are also included in our specifications.

## III Data

### 3.1 Data sources and variable definitions

The data of macroeconomic variables (GDP, capital stock, investment, population size, employment rate, import and export, average working hours, labor share, etc.) are retrieved from PWT 9.0. Educational attainment is defined as the average years of primary, secondary, and tertiary schooling among the population above 15 years old, and the data are from Barro and Lee (2013) [53]. Inflation is the yearly percentage change in price using year-average CPI; we retrieve the inflation and financial development indices from the IMF. We obtain civil liberty and freedom dummy data from Freedom House. As the usual strategy to reduce frequent data variations, we use five-year average data. The statistical summary of the data is presented in S1 Table in S1 Appendix.

We collect the income distribution panel data from WIID3.4 of UNU-WIDER (https://www.wider.unu.edu/project/wiid-world-income-inequality-database). The panel data are consistent and comparable because the data are chosen with consistent statistical units, a few of which are estimated by the methodology that Shao (2017) [54] uses; for instance, the income definition is disposable income, the income unit is per household per capita, and the data survey is nationwide and covers all ages in the population. We mainly consider 7 statistical measurements of income distribution: the Gini coefficient, the bottom (B20) and top (T20) quintiles of income share, the income share of the 60% middle incomes (MID, total income share of the three middle quintiles), the income share below the median (MES), the income share below the national mean income (MIS), and the share of the population whose individual income is below the national mean income (MPS. For a more detailed discussion about MPS and MIS, refer to Shao, 2017 [54]).

The variable definitions are as follows (subscripts for country and year are dropped for conciseness):

$GDP = log\ (cgdpo/pop)$, where $cgdpo$ is output-side real GDP at current PPPs (in 2011 US dollars), and pop is the total population size in a country in a particular year.

HR = log(avh), where $avh$ is the average annual hours worked by persons engaged in a country.

Emp = emp/pop, where $emp$ is the total number of persons engaged in a job during a year in a country.

$PG$ is the population growth rate, and $pi$ is the inflation rate. $Frdm$ is an indicator dummy for freedom, and $CL$ is the civil liberty index; both are from the Freedom House (https://freedomhouse.org/report/freedom-world/freedom-world-2018). $FDI$ is the financial development index from the IMF (http://data.imf.org/?sk=F8032E80-B36C-43B1-AC26-493C5B1CD33B).

$Ksh = ck/cgdpo$, where $ck$ is capital stock at current PPPs (in 2011 US dollars).

$Ish = csh\_i$, which is the share of gross capital formation in $cgdpo$ at current PPPs.

$Gsh = csh\_g$, which is the share of government consumption in in $cgdpo$ at current PPPs.

*Opnsh = csh_x-csh_m*, which is the ratio definition for openness.

The level definitions for capital stock, investment, government spending, exports and imports taking log per-capita form are as follows:

$$\text{K} = \log\left(\frac{ck}{pop}\right), \text{I} = \log\left(csh_i * \frac{cgdpo}{pop}\right), \text{G} = \log\left(csh_g * \frac{cgdpo}{pop}\right),$$

$$\text{XT} = \log(csh\_x * cgdpo/pop), \text{MT} = \log(csh\_m * cgdpo/pop),$$

$$Opnsh = csh_x - csh_m, Open = \log(Opnsh * cgdpo/pop).$$

$csh_x(csh_m)$ is the share of exports (imports) in output-side real GDP (cgdpo) at the current domestic price. $csh_m$ takes a negative sign in PWT9.0, *Open* is the log of per capita total trade and *Opnsh* is the share of total international trade in output-side real GDP. The level definition of these variables is needed to partition GDP into explained and unexplained proportions, and their ratio definitions are needed to estimate the statistical measures of income distribution to avoid high multicollinearity.

We do not use a summary index to measure human capital in a country. However, we include the average years of primary (pry), secondary (sey), and tertiary (tey) education to indicate different levels of human capital because these three levels of education may not affect income distribution in the same fashion. We also use interactive terms to describe the nonlinear effects of inflation, civil liberty, and openness with development (GDP).

## 3.2 Data summary of the statistical measures of income distribution

A typical property of panel data is that within-variation and between-variation are quite different. Li, Squire and Zou (1998) [2] identify differences between within-variation and between-variation in inequality, but their specifications take no action to identify this difference. The within-country variation (standard deviation) is only approximately one-third as large as the between-country variation for the Gini coefficient, and it is approximately one-half as large as that for the MPS and MIS. These findings are shown in Table 1, which summarizes the data for the Gini coefficient, B20, T20, MES, MID, MPS, MIS, and GDP.

The other variables also show much larger between-variation than within-variation; one exception is that the within-variation of inflation is approximately 3 times greater than the between-variation. The data summary of all other variables is relegated to S1 Table in S1 Appendix. This property of panel data is very informative in forming specifications of inequality variation and choosing estimators. That is, both between- and within-variations of inequality must be properly identified, and the same consideration applies to all other explanatory variables. We take two actions to address this issue. First, we apply the simple dynamic LSDV estimator to the panel data, which ensures good efficiency of the regressions and is said to outperform GMM and system GMM (Moral-Benito, 2013) [4]. Second, we isolate the country and time components from each variable so that the between-effects and within-effects in each variable are isolated and multicollinearity can be dramatically reduced among all explanatory variables.

Table 2 below shows the bilateral correlation among the variables GDP, GDP², investment (I), employment (Emp), tertiary education (tey), average working hours (HR), Open and Opnsh.

Table 2 shows that the bilateral correlation among the variables is very strong and that the correlation of Open with other variables is much larger than that of Opnsh, and it is similar for the definitions of capital stock, investment and government spending, which is why we prefer the ratio to the log-level definition for these variables in this study. S2 Table in S1 Appendix shows the countries and years for the data of the Gini coefficient.

**Table 1. Data summary of income inequality and GDP.**

| Variable | | Mean | Std. Dev. | Min | Max | Observations |
|---|---|---|---|---|---|---|
| Gini | overall | 0.383 | 0.112 | 0.199 | 0.840 | N = 488 |
| | between | | 0.104 | 0.220 | 0.655 | n = 83 |
| | within | | 0.037 | 0.151 | 0.639 | T-bar = 5.88 |
| B20 | overall | 0.068 | 0.023 | 0.017 | 0.119 | N = 414 |
| | between | | 0.021 | 0.030 | 0.106 | n = 82 |
| | within | | 0.009 | 0.026 | 0.100 | T-bar = 5.05 |
| T20 | overall | 0.441 | 0.084 | 0.202 | 0.651 | N = 414 |
| | between | | 0.076 | 0.326 | 0.608 | n = 82 |
| | within | | 0.027 | 0.266 | 0.606 | T-bar = 5.05 |
| MES | overall | 0.255 | 0.061 | 0.121 | 0.363 | N = 414 |
| | between | | 0.055 | 0.146 | 0.347 | n = 82 |
| | within | | 0.020 | 0.156 | 0.321 | T-bar = 5.05 |
| MID | overall | 0.491 | 0.062 | 0.324 | 0.583 | N = 414 |
| | between | | 0.057 | 0.361 | 0.568 | n = 82 |
| | within | | 0.020 | 0.350 | 0.557 | T-bar = 5.05 |
| MIS | overall | 0.373 | 0.042 | 0.188 | 0.470 | N = 488 |
| | between | | 0.040 | 0.233 | 0.450 | n = 83 |
| | within | | 0.020 | 0.271 | 0.437 | T-bar = 5.88 |
| MPS | overall | 0.645 | 0.051 | 0.534 | 0.821 | N = 488 |
| | between | | 0.046 | 0.573 | 0.754 | n = 83 |
| | within | | 0.018 | 0.581 | 0.757 | T-bar = 5.88 |
| GDP | overall | 9.383 | 0.993 | 6.313 | 11.284 | N = 449 |
| | between | | 0.931 | 6.681 | 10.799 | n = 79 |
| | within | | 0.368 | 7.882 | 10.553 | T-bar = 5.68 |

Note: N denotes the number of observations; n denotes the number of countries. T-bar is the average years a variable is observed in a country. The data are 5-year averaged.

### 3.3 Transfer panel data into quasi-cross-sectional data

Bilateral correlation likely leads to high multicollinearity in estimation. Strong multicollinearity may result in biased and inconsistent estimation when relevant and correlated variables are omitted. To reduce the bilateral correlation among the covariates in the panel data, we run

**Table 2. Bilateral correlations among selected variables in the original data.**

| | GDP | GDP$^2$ | Ish | Emp | HR | CL | tey | Open | Opnsh |
|---|---|---|---|---|---|---|---|---|---|
| GDP | 1.00 | | | | | | | | |
| GDP$^2$ | **1.00** | 1.00 | | | | | | | |
| Ish | **0.49** | **0.49** | 1.00 | | | | | | |
| Emp | **0.60** | **0.61** | 0.42 | 1.00 | | | | | |
| HR | **-0.59** | **-0.59** | -0.15 | -0.47 | 1.00 | | | | |
| CL | **-0.68** | **-0.68** | -0.28 | -0.37 | **0.50** | 1.00 | | | |
| tey | **0.59** | **0.59** | 0.17 | 0.40 | -0.25 | -0.33 | 1.00 | | |
| Open | **0.93** | **0.93** | **0.54** | **0.56** | **-0.54** | **-0.65** | **0.52** | 1.00 | |
| Opnsh | **0.57** | **0.58** | 0.44 | 0.39 | -0.33 | -0.39 | 0.32 | **0.78** | 1.00 |

Note: N = 274. The data are five-year averaged.

OLS regressions for each variable $x_{it}$ on the country dummies and year dummy (or level year for the trended effect); which is the methodology of detrending and deseasonalizing time series introduced in Wooldridge [5]. We save the residuals as the new data of the variable, denoted by $nx_{it}$, which is the proportion of the variable that is not explained by country and time. The country-fixed and time-effect components of explanatory variables are often the main channels of high multicollinearity; the bilateral correlation among the new data $nx_{it}$ of the variables will be much smaller than that among the original data. When we run regressions with the new data $nx_{it}$, multicollinearity is dramatically reduced or even removed so that the imprecision and inconsistent issues are effectively reduced or even removed.

We run OLS regressions for GDP on all the new variables, including current and 5-year lagged inequality terms (MPS, MIS, MID, T20, and Gini), country dummies, and a form (fixed or trended) of the time variable. Then, we save the predictions and residuals, which form explained GDP and unexplained GDP, denoted by GDP$e_{it}$ and $e_{it}$, respectively. To remove the country and year components of GDP$e_{it}$, we also run OLS regression for it on country dummies and year (dummy or level) again and save the residuals, which are the new $GDP_{it}$, $nGDP_{it}$.

The above regression equations are as follows:

$$x_{it} = \alpha_0 + \alpha_1 i.\text{country} + \alpha_2 t(year) + nx_{it}, \tag{3.3.1}$$

$$\text{GDP}_{it} = X_{it}\beta + \alpha_1 i.\text{country} + \alpha_2 t(year) + e_{it}, \tag{3.3.2}$$

$$\widehat{GDP}_{it} = nX_{it}\widehat{\beta} + \widehat{\alpha}_1 i.\text{country} + \widehat{\alpha}_2 t(year) \tag{3.3.3}$$

$$\widehat{GDP}_{it} = \alpha_0 + \alpha_1 i.\text{country} + \alpha_2 t(year) + nGDP_{it}, \tag{3.3.4}$$

where the residuals of (3.3.1), $nx_{it}$, generate the new (quasi-cross-sectional) data of $x_{it}$; the residuals of (3.3.2), $\widehat{e}_{it}$, form unexplained GDP, and its predicted values, $\widehat{GDP}_{it}$, are explained GDP; $nX_{it}$ is the vector of all new variables $nx_{it}$; $\beta$ and $\alpha$ ($\widehat{\beta}$ and $\widehat{\alpha}$) are (estimated) coefficients; subscript $i$ is a country index; $t(year)$ is a function of time $t$, either fixed or trended in time; $\alpha_2$ * $i.year$, or $\alpha_2$ * $year$, $i.year$ and $i.country$ are year and country dummies, respectively; and the residuals of (3.3.4), $n\widehat{GDP}_{it}$, generate the new data (quasi-cross-sectional) of $\widehat{GDP}_{it}$. Note that to simplify the analysis, constant trend $\alpha_2$ for all countries is implied for $t(year)$ taking the level form $\alpha_2$ * $year$.

The vector $X_{it}$ in (3.3.2) includes the current and 5-year-lagged Gini coefficient, MIS, MPS, MID and T20. Capital stock, investment, exports and imports are in level definitions, but we use the ratio definition later for these variables when we run regressions of the 7 statistical measures of income distribution to avoid high multicollinearity in the regressions of income distribution statistics.

Since the new data (quasi-cross-sectional) are defined by residuals of the OLS regressions (3.3.1), all variables with the new data, including $n\widehat{GDP}_{it}$, $n\widehat{GDP}_{it}^2$, $e_{it}$ and $e_{it}^2$, are centered at zero. Furthermore, any bilateral correlations among $nGDP_{it}$, $nGDP_{it}^2$, $e_{it}$, and $e_{it}^2$ are near zero. $e_{it}$, and $e_{it}^2$ are uncorrelated with all other explanatory variables, and all new variables $nx_{it}$ including $n\widehat{GDP}_{it}$, $n\widehat{GDP}_{it}^2$, $e_{it}$, and $e_{it}^2$ are not correlated with country dummies and time year.

Table 3 below summarizes the statistics of the selected variables corresponding to Table 1, which have been transferred into quasi-cross-sectional data. As observed, the standard deviation of the new variable is much smaller than that of the corresponding original variable

**Table 3. Summary statistics of selected new variables (quasi-cross-sectional data).**

| Variable | Mean | Trended time | | | Fixed-effect time | | | N(n) |
|---|---|---|---|---|---|---|---|---|
| | | Std.Dev. | Min | Max | Std.Dev. | Min | Max | |
| nGini | 0.000 | 0.037 | -0.225 | 0.245 | 0.036 | -0.219 | 0.233 | 488 (5.88) |
| nB20 | 0.000 | 0.009 | -0.042 | 0.035 | 0.009 | -0.038 | 0.030 | 414(5.05) |
| nT20 | 0.000 | 0.027 | -0.174 | 0.167 | 0.026 | -0.182 | 0.156 | 414(5.05) |
| nMES | 0.000 | 0.020 | -0.101 | 0.068 | 0.019 | -0.091 | 0.057 | 414(5.05) |
| nMID | 0.000 | 0.020 | -0.141 | 0.067 | 0.019 | -0.128 | 0.055 | 414(5.05) |
| nMIS | 0.000 | 0.020 | -0.102 | 0.064 | 0.020 | -0.099 | 0.062 | 488 (5.88) |
| nMPS | 0.000 | 0.018 | -0.063 | 0.115 | 0.017 | -0.051 | 0.122 | 488 (5.88) |
| nGDP | 0.000 | 0.089 | -0.298 | 0.330 | 0.087 | -0.290 | 0.296 | 180(3.6) |

Note: N denotes the number of observations; n denotes the average years a variable is observed in a country.

because the between-country and overtime variations, which are generally large, have been removed from the original data.

Table 4 below shows the bilateral correlations among the new variables corresponding to those in Table 2. The table shows that the bilateral correlations with the new (quasi-cross sectional) data have become much smaller than those with the original panel data in Table 2; for instance, the correlation coefficient between GDP and $GDP^2$ is 1, but it is -0.007 between nGDP and $nGDP^2$; this is because the panel data to be used in Table 4 have become the quasi-cross sectional data by removing the country and time effects from the original panel data to be used in Table 2, which removes the correlation effects of country and time in the original panel data and often changes the direction of the data as well. Note that what we care about here is the small size of the bilateral correlation coefficient in Table 4 so that the explanatory power (or say multicollinearity) is small too when we look at the regression of one variable on all other variables using the quasi-cross section data.

## 3.4 An application of the quasi-cross-sectional data on inequality measures

The Gini coefficient is closely related to MPS and MIS. We run OLS regressions of the Gini coefficient on MPS, MIS and their product MDS with trended time effects on the quasi-cross-sectional data. We use three estimators, POLS, fixed effects (FE) and LSDV, with robust errors. The POLS estimator explains the correlation between countries, and the FE and LSDV

**Table 4. Bilateral correlation among selected new variables.**

| | e | nGDP | nGDP² | nIsh | nEmp | nHR | nCL | ntey | nOpen |
|---|---|---|---|---|---|---|---|---|---|
| e | 1.000 | | | | | | | | |
| nGDP | 0.000 | 1.000 | | | | | | | |
| nGDP² | 0.066 | -0.007 | 1.000 | | | | | | |
| nIsh | -0.265 | 0.256 | -0.120 | 1.000 | | | | | |
| nEmp | 0.000 | 0.284 | 0.073 | 0.302 | 1.000 | | | | |
| nHR | 0.000 | -0.060 | -0.026 | 0.283 | -0.104 | 1.000 | | | |
| nCL | 0.000 | 0.190 | 0.024 | 0.038 | 0.224 | -0.026 | 1.000 | | |
| ntey | 0.000 | 0.071 | -0.019 | -0.044 | 0.168 | -0.189 | 0.091 | 1.000 | |
| nOpen | 0.002 | 0.359 | -0.056 | 0.480 | 0.103 | 0.115 | -0.246 | -0.032 | 1.000 |
| nOpnsh | -0.177 | -0.013 | -0.070 | 0.215 | -0.080 | 0.072 | -0.182 | 0.072 | 0.551 |

Note: N = 221. All variables (subscripts of country and year are omitted) in the table are generated by the residuals $nx_{it}$ in Eqs (3.3.1), (3.3.2) and (3.3.4). $nGDP^2 = (nGDP)^2$.

**Table 5. Regression of the Gini coefficient on mean-income shares with trended time.**

| Regressions with the quasi-cross-sectional data | | | | | | |
|---|---|---|---|---|---|---|
| Variable | POLS | | FE | | LSDV | |
| nMPS | 1.246*** | **1.250***** | 1.244*** | 1.250*** | 1.330*** | **1.331***** |
| nMIS | -0.965*** | **-0.962***** | -0.966*** | -0.962*** | -1.278*** | **-1.280***** |
| nMDS | 2.584 | | 3.492 | | 0.252 | |
| _cons | -0.000 | -0.000 | -0.000 | -0.000*** | 0.450*** | **0.450***** |
| N | 488 | 488 | 488 | 488 | 488 | 488 |
| R²_a | 0.527 | **0.527** | 0.528 | 0.527 | 0.997 | **0.997** |
| Regressions with the original panel data | | | | | | |
| MPS | 0.528*** | 1.327*** | 0.536 | 1.276*** | 0.475*** | **1.331***** |
| MIS | -2.632*** | -1.237*** | -2.343** | -0.963*** | -2.631*** | **-1.280***** |
| MDS | 2.196*** | | 2.085 | | 2.213*** | |
| _cons | 0.494*** | -0.016 | 0.411 | -0.081 | 0.527*** | -0.001 |
| N | 488 | 488 | 488 | 488 | 488 | 488 |
| R²_a | 0.991 | 0.99 | 0.53 | 0.526 | 0.997 | 0.997 |

Note: The data are five-year averages. $MDS = MPS * MIS$, $nMDS = nMPS * nMIS$. We perform a serial test using pantest2 in Stata for the FE estimation, which suggests rejection of the null of serial correlation. The dependent variables for the POLS and FE estimators are nGini on the quasi-cross-sectional data and Gini for the other regressions. The time variable year is demeaned at 2000 to have robust intercepts and included in LSDV regressions.

\* $p < 0.05$

\*\* $p < 0.01$

\*\*\* $p < 0.00$.

estimators explain the correlation within countries. For comparison, we also run the same regressions with the original panel data. Table 5 below shows the results. The results with time dummies are similar and relegated to S3 Table in S1 Appendix.

We observe the following results from Table 5 and S3 Table in S1 Appendix:

- The estimates of nMIS and nMPS are robust for the two specifications (one includes nMDS; the other does not) in the three estimators with the quasi-cross-sectional data but not in those with the original panel data.

- The LSDV estimates of linear terms are the same as the two data sets (see the last column on the right). They are robust with the quasi-cross-sectional data (see the last two columns on the right in the upper four rows) but are not robust with the original panel data (see the last two columns on the right in the lower four rows).

- POLS estimates are robust with the quasi-cross-sectional data (see the first two columns on the left in the upper four rows) but not robust with the original panel data (see the first two columns in the lower four rows).

Therefore, the Gini coefficient between countries can be empirically described as **1.25**$MPS$ **−0.962**$MIS$ from the pooled OLS estimation, and it can be approximately noted within countries as $1.331MPS - 1.28MIS + 0.45$ from the LSDV estimation, both of which use quasi-cross-sectional data. Note that FE and LSDV do not have the same estimates because robust errors are used and year is included in LSDV but not in FE.

## IV Literature replications

We replicate the estimation of three widely cited papers on this topic, namely, Barro (2000, 2008) [1, 17], Li et al. (1998) [2] and Milanovic (2005) [24], and show how their results are

biased and not robust due to multicollinearity of panel data. In the next subsection, we show that LSDV and POLS estimations become robust estimators with the quasi-cross-sectional data.

There are two differences between our data and theirs. One is that we choose a comparable and consistent (disposable income) Gini coefficient, in which we estimate a few observations from the Gini coefficient of other income definitions by assuming constant linear correlation over time between the Gini coefficients of disposable income and other income definitions in a country (Shao, 2017) [54]. In contrast, the other three studies include inequality data with different income definitions and statistical units and use dummy variables to denote these different statistical dimensions. The other difference is in the treatment of data frequency. Barro (2000, 2008) [1, 17] uses 10-year average data, while Li et al. (1998) [2], Milanovic (2005) [24], and this paper use 5-year average data.

## 4.1 Barro's (2000) specifications

We replicate the six specifications (Table 6 Part I, page 23, Barro, 2000 [1]) that account for the Gini coefficient (similar results for Part II of FE estimation are not listed here to save space). The time period of the data in Barro (2000) [1] is from 1960 to 1990, and Barro (2009) [17] confirms the results by extending the data to 2000. Table 6 below shows the replication results of the seemingly unrelated regression (SUR) estimation on our panel data.

The replication results show significant and negative effects on quadratic GDP in all regressions. The average turning point (8.24) of the estimated ln(GDP) is smaller than the mean (9.38) of the sample ln(GDP), and the standard deviation is 0.218, which is similar to the result in Barro (2000, 2009) [1, 17], and so are the estimates of educational attainment. The small differences might be caused by the data averages and specifications. Our data set size is much larger, and we do not include dummies for income definitions because we estimate the Gini

**Table 6. Replication results for Barro (2000) using the original panel data.**

| Variable | (1) | (2) | (3) | (4) | (5) | (6) | VIF |
|---|---|---|---|---|---|---|---|
| GDP | 0.235** | 0.377*** | 0.476*** | 0.454*** | 0.372*** | 0.173* | 686 |
| GDP$^2$ | -0.014*** | -0.023*** | -0.028*** | -0.027*** | -0.023*** | -0.011* | 734 |
| tey | | 0.071*** | 0.064*** | 0.057*** | 0.071*** | 0.073*** | 2.69 |
| sey | | 0.013** | 0.016*** | 0.018*** | 0.013** | 0.009 | 4.03 |
| pry | | -0.001 | 0.002 | 0.002 | -0.001 | 0.003 | 2.7 |
| Frdm | | | -0.026** | | | | 3.36 |
| CL | | | | 0.008* | | | 4.88 |
| Opnsh | | | | | -0.002 | 0.182 | 248 |
| Opnsh*GDP | | | | | | -0.017 | 274 |
| N | 347 | 347 | 321 | 296 | 347 | 347 | |
| R$^2$ | 0.680 | 0.725 | 0.752 | 0.780 | 0.725 | 0.909 | |

Note: The dependent variable is the Gini coefficient, and the estimator is seemingly unrelated regression (SUR). The years of data are 1990, 1995, 2000, 2005, and 2010. Coefficients for year dummies are not listed here to save space. The variable *Opnsh* is defined as the ratio of imports plus exports to GDP. The dummy variables for country area are advanced economies, Europe and central Asia, South Asia, East Asia and the Pacific, Latin America and the Caribbean, Sub-Saharan Africa, and the Middle East and North Africa. Taking the same approach as Barro [1], area dummies are included in columns (1)-(5), and column 6 includes country dummies. Column VIF shows the variation inflation factor for the OLS regression of the Gini coefficient on all regressors including area dummies.

* $p < 0.05$

** $p < 0.01$

*** $p < 0.001$.

**Table 7. Replication results for Barro (2000) with the quasi-cross-sectional data.**

| Variable | (1) | (2) | (3) | (4) | (5) | (6) | VIF |
|---|---|---|---|---|---|---|---|
| nGDP | 0.008 | -0.001 | -0.003 | 0.000 | -0.001 | -0.009 | 1.08 |
| nGDP$^2$ | -0.058 | -0.055 | -0.040 | -0.054 | -0.061 | -0.813*** | 1.12 |
| ntey | | 0.088 | 0.085 | 0.088 | 0.088 | 0.019 | 1.18 |
| nsey | | 0.001 | 0.001 | 0.001 | 0.001 | 0.018* | 1.15 |
| npry | | -0.006 | -0.005 | -0.006 | -0.006 | -0.024 | 1.41 |
| nFrdm | | | -0.013 | | | | 1.55 |
| nCL | | | | -0.001 | | | 1.62 |
| nOpnsh | | | | | -0.009 | -0.026 | 1.1 |
| nOpnsh*nGDP | | | | | | -0.068 | 1.14 |
| N | 180 | 180 | 180 | 180 | 180 | 180 | |
| R$^2$ | 0.779 | 0.784 | 0.784 | 0.784 | 0.784 | 0.938 | |

Note: Dependent variables are the Gini coefficient with the original panel data, and other variables are from the quasi-cross-sectional data.

nOpshGDP = nOpnsh*nGDP. Area dummies are included in all regressions except column 6, which includes country dummies. The variable definitions refer to Eqs (3.3.1) ~ (3.3.4), and the time variable is in fixed-effect form. The OLS regression in column VIF includes all variables and area and year dummies.

* p<0.05

** p<0.01

*** p<0.001.

coefficient to make the data consistent and comparable regarding definitions and statistical issues. The VIF column shows very large values that reveal a high possibility of omitted variable bias. We list the VIF values for only one regression that includes all regressors since the VIF values are generally not much different in other regressions that drop some of the regressors; the same reason applies to the VIF values in other tables.

We argue that Barro's results are misled by high multicollinearity, and the results are biased and not robust. The multicollinearity in the estimations is caused by both structural terms, for instance, GDP and GDP$^2$, and data issues, as all variables show country-fixed and year-fixed (or trended) effects.

We use the quasi-cross-sectional data obtained from the estimations by Eqs (3.3.1) ~ (3.3.4) to run the same regressions on Barro's specifications. The results are listed in Table 7 below. The small values in column VIF show that multicollinearity is no longer a concern in the regressions.

Table 7 shows that we do not find evidence for the Kuznets hypothesis across countries. The SUR estimator assumes that the Gini coefficient must be independent across countries. This assumption is rejected by the Breusch-Pagan test after the estimation, which is not reported in the table to save space. Note that the results are still similar if the time variable is assumed to be trended in generating the quasi-cross-sectional data. Therefore, Barro's estimation results are not robust to quasi-cross-sectional data.

## 4.2 Li, Squire and Zou's (1998) specifications

Li, Squire and Zou (1998) [2] (LSZ for short) explain income inequality by a reduced form of initial mean level of secondary education, financial development, civil liberty, and initial land inequality. Table 8 below shows the replication results, which refer to the base and IV regressions in Table 6 and columns 1, 2, 3, and 5 in Table 8 of LSZ. We can see that the estimation results of civil liberty in all regressions and of secondary education and financial development in the regressions of columns OLS and IV(1) are the same as LSZ's estimates regarding their

**Table 8. Replication results of selected regressions in LSZ (1998) with IV estimation.**

| Variable | OLS | IV(1) | IV(2) | IV(3) | IV(4) | VIF |
|---|---|---|---|---|---|---|
| L5.sey | -0.020*** | -0.022* | -0.021 | -0.021 | -0.022 | 2.09 |
| CL | 0.044*** | 0.030** | 0.028** | 0.029** | 0.029** | 2.03 |
| FDI | -0.080** | -0.167** | -0.143 | -0.131 | -0.107 | 2.59 |
| L5.GDP | | | -0.01 | -0.006 | -0.004 | 4.54 |
| Xsh | | | | -0.036 | -0.026 | 1.42 |
| Ish | | | | | -0.208 | 1.38 |
| _cons | 0.385*** | 0.465*** | 0.551** | 0.515* | 0.537** | |
| N | 240 | 227 | 133 | 133 | 133 | |
| # of countries | 62 | 66 | 44 | 44 | 44 | |

Note: The dependent variable is the Gini coefficient for all regressions. Xsh is the ratio of exports to GDP, and Ish is the share of investment within GDP. Column OLS uses robust OLS regression, while the other columns use IV regression. The instrument is L5.FDI for FDI, which is the same as in LSZ. The IV regressions are for between-effects. The OLS regression in column VIF includes all related variables.

* p<0.05

** p<0.01

*** p<0.001.

signs and significance levels. Note that our financial development index (FDI) is a more comprehensive measure than the ratio of M2 to GDP, which LSZ uses.

We do not have data of land inequality, but our robust OLS regression without initial land inequality provides very similar results regarding the signs and significance levels of civil liberty and financial development. The estimation of initial secondary education shows the same sign as LSZ, but the significance level is only 5% (see the OLS column in Table 9 below). The replication is not exactly the same as LSZ since the estimation for financial development in the IV replications is not significant, but it is sufficient to replicate the estimates on civil liberty and secondary education.

Table 9 below lists the replication results of LSZ [2] with our quasi-cross-sectional data. All of the significant results in Table 8 have now disappeared or reduced, and the effect of investment

**Table 9. Replication of LSZ [2] with quasi-cross-sectional data.**

| Variable | OLS | IV(1) | IV(2) | IV(3) | IV(4) | VIF |
|---|---|---|---|---|---|---|
| L5.nsey | -0.010* | -0.008 | 0.006 | 0.004 | 0.026 | 1.01 |
| nCL | 0.001 | 0.021 | 0.001 | 0.002 | 0.009 | 1.02 |
| nFDI | -0.022 | -0.365* | 0.202 | 0.196 | 0.168 | 1.05 |
| L5.nGDP | | | -0.051 | -0.051 | 0.037 | 1.05 |
| nXsh | | | | 0.02 | -0.002 | 1.02 |
| nIsh | | | | | 0.369* | 1.06 |
| _cons | -0.003** | -0.004 | -0.001 | -0.001 | -0.002 | |
| N | 240 | 227 | 133 | 133 | 133 | |
| # of countries | 62 | 66 | 44 | 44 | 44 | |

Note: The dependent variable for all regressions is nGini, the Gini coefficient of quasi-cross-sectional data. The definitions for Xsh and Ish are the same as those in Table 8. Column OLS uses robust OLS regression. The IV regressions are for between-effects. Variable definitions refer to Eqs (3.3.1) ~ (3.3.4), and time is in fixed-effect form for transferring the panel data to quasi-cross-sectional data. The OLS regression in column VIF includes all related variables.

* p<0.05

** p<0.01

*** p<0.001.

has become significantly positive but it was insignificant and negative in the LSZ regressions. The very small values in the VIF column show that multicollinearity is no longer a concern in the regressions. The results are still similar if the time variable is assumed to be trended. Therefore, LSZ's estimation results are not robust to the specifications or quasi-cross-sectional data.

## 4.3 Milanovic (2005)

Milanovic (2005) [24] studies how openness and direct foreign investment affect income distribution within a country. He finds that poor people in low-income countries have less income share when the countries are more open to trade; however, as national incomes rise, the poor and middle class benefit more than the rich. We replicate Milanovic (2005) with similar variables, but we use total investment to replace FDI and the freedom dummy to replace democracy due to data availability in our data set.

Taking the usual approach in the literature (Milanovic, 2005; Barro, 2000; Ravallion, 2001; Dollar and Kraay, 2002) [1, 24–26], using the interaction of openness and income to denote their nonlinear relationship, we also consider the interactive term between income and investment share within output. The replication results are shown in Table 10 below.

**Table 10. Replication of Milanovic (2005).**

| Variable | D1 | D2 | D3 | D4 | D5 | VIF |
|---|---|---|---|---|---|---|
| L5.Opnsh | -0.151** | -0.128 | -0.158** | -0.093 | -0.149* | 13 |
| L5.Gsh | 0.067*** | 0.082*** | 0.083*** | 0.076*** | 0.073*** | 1.2 |
| GDP | -0.895* | -0.451 | -0.6 | -0.118 | -0.486 | 20 |
| Opnsh*GDP | 0.015** | 0.013 | 0.016** | 0.009 | 0.015* | 14 |
| FDI | 0.192 | 0.043 | -0.231 | -0.369 | -0.314 | 3.22 |
| Ish | -0.112 | -0.136 | -0.151 | -0.141 | -0.14 | 109 |
| Ish*GDP | 0.014 | 0.015 | 0.017 | 0.016 | 0.017 | 166 |
| Frdm | 0.676** | 0.625 | 0.907*** | 0.652 | 0.910*** | 1.53 |
| pi | -0.054*** | -0.061** | -0.084*** | -0.075*** | -0.096*** | 1.04 |
| N | 234 | 225 | 234 | 225 | 234 | |
| Adj_$R^2$ | 0.249 | 0.413 | 0.375 | 0.489 | 0.371 | |
| Variable | D6 | D7 | D8 | D9 | D10 | VIF |
| L5.Opnsh | -0.037 | -0.120* | 0.078 | -0.091 | 0.267 | 13 |
| L5.Gsh | 0.054*** | 0.045** | 0.005 | -0.045* | -0.417*** | 1.19 |
| GDP | 0.161 | -0.228 | 0.559 | -1.625* | -0.006 | 20 |
| Opnsh*GDP | 0.004 | 0.012* | -0.008 | 0.008 | -0.027 | 14 |
| FDI | -0.392 | -0.305 | -0.29 | 0.863 | -0.92 | 3.24 |
| Ish | -0.125 | -0.062 | -0.061 | -0.196 | 0.939 | 110 |
| Ish*GDP | 0.015 | 0.01 | 0.008 | 0.026 | -0.114 | 166 |
| Frdm | 0.468 | 0.692*** | 0.099 | 0.842* | -3.316 | 1.66 |
| pi | -0.069*** | -0.088*** | -0.029* | -0.040* | 0.428*** | 1.04 |
| N | 225 | 234 | 225 | 234 | 225 | |
| Adj_$R^2$ | 0.522 | 0.227 | 0.158 | 0.103 | 0.503 | |

Note: All variables are 5-year averaged. The dependent variables (D1~D10) are 5-year averaged decile income shares. Opnsh, Gsh, and Ish are the ratios of total trade, government spending, and total investment to GDP in PPP 2011 international dollars, respectively. The estimator is the instrumental GMM (ivreg2 in Stata), and Opnsh and Gsh are instrumented by their 5-year lags and population. Column VIF is for the OLS regression with dependent variable D5. Column VIF is for the OLS regression of dependent variable D10.

* p<0.05

** p<0.01

*** p<0.001.

**Table 11. Replication of Milanovic (2005) with the quasi-cross-sectional data.**

| Variable | nD1 | nD2 | nD3 | nD4 | nD5 | VIF |
|---|---|---|---|---|---|---|
| L5.nOpnsh | -0.004 | -0.006 | -0.004 | 0.001 | 0.003 | 1.1 |
| L5.nGsh | -0.062* | -0.07 | -0.063 | -0.053 | -0.046 | 1.13 |
| nGDP | 0.008* | 0.006 | 0.005 | 0.003 | 0.004 | 1.18 |
| nOpnsh*nGDP | 0.007 | 0.021 | 0.024 | 0.038 | 0.047* | 1.05 |
| nFdi | -0.004 | -0.005 | -0.004 | -0.001 | -0.002 | 1.07 |
| nIsh | -0.035** | -0.039** | -0.037** | -0.028** | -0.024** | 1.2 |
| nIsh*nGDP | -0.058 | -0.084 | -0.082 | -0.122 | -0.128 | 1.13 |
| nFrdm | -0.001 | -0.001 | -0.001 | 0 | 0 | 1.06 |
| npi | -0.000** | -0.000* | -0.000** | -0.000** | -0.000*** | 1.16 |
| N | 173 | 173 | 173 | 173 | 173 | |
| Adj_R$^2$ | -0.009 | -0.042 | -0.059 | -0.07 | -0.071 | |
| Variable | nD6 | nD7 | nD8 | nD9 | nD10 | VIF |
| nOpnsh | 0.003 | 0.004 | 0.002 | 0 | -0.022 | 1.1 |
| nGsh | -0.039 | -0.034 | -0.007 | -0.009 | 0.358 | 1.13 |
| nGDP | 0.006 | 0.003 | 0.002 | -0.013* | -0.026 | 1.18 |
| nOpnsh*nGDP | 0.067** | 0.044* | 0.042*** | -0.047 | -0.302 | 1.05 |
| nFdi | 0.003 | -0.007 | -0.003 | -0.011 | -0.051 | 1.07 |
| nIsh | -0.018* | -0.007 | 0.005 | 0.054* | 0.158** | 1.2 |
| nIsh*nGDP | -0.13 | -0.096 | -0.003 | 0.153 | 0.712 | 1.13 |
| nFrdm | 0 | 0 | 0.002* | 0.002 | 0.002 | 1.06 |
| npi | -0.000*** | -0.000*** | -0.000* | 0 | 0.002** | 1.16 |
| N | 173 | 173 | 173 | 173 | 173 | |
| Adj_R$^2$ | -0.045 | -0.092 | -0.011 | 0.038 | -0.06 | |

Note: Dependent variables are the quasi-cross-sectional data of 5-year averaged decile income shares. The estimator is the instrumental GMM (ivreg2 in Stata), and Opnsh and Gsh are instrumented by their 5-period lags and population. Variable definitions refer to Eqs (3.3.1) ~ (3.3.4), and time is in fixed-effect form. Column VIF is for the OLS regression of dependent variable nD5. Column VIF is for the OLS regression with dependent variable nD10.

\* p<0.05

\*\* p<0.01

\*\*\* p<0.001. The year variable takes the fixed-effect form for all variables and regressions.

The replication also shows some results similar to those of Molanovic's Tables 3 and 4. Openness significantly decreases the income shares of the bottom first and third deciles in low-income countries (Opnsh*GDP). The effects of openness on the middle-income deciles are indeed positive, but the effects are not significant for the bottom second or fourth deciles. Additionally, the effects of the level of openness are negative and significant for the bottom 1[st], 3[rd], 5[th] and 7[th] deciles. Government spending also shows similar results to Milanovic's results.

We run the same specifications with our quasi-cross-sectional data, and the results are listed in Table 11 below.

Table 11 shows that after multicollinearity has been reduced in the same specification the instrumental GMM estimation does not present significant effects of openness or government spending on the bottom deciles in low-income countries (OpshGDP). The negative R$^2$ indicates poor power of the IV GMM estimator because endogeneity is no longer an issue in the data, and R$^2$ becomes reasonably positive with LSDV and POLS. Note that the results are still similar if the time variable is assumed to be trended. Therefore, Milanovic's estimation results are not robust to the quasi-cross-sectional data.

## V Robust determinants of income distribution with the quasi-cross-sectional data

We use static and dynamic specifications with trend and fixed-effect time forms to explore robust determinants of income inequality across countries by POLS and within countries by LSDV on the quasi-cross-sectional data.

Many studies (e.g., Rodriguez-Pose and Tselios, 2009; Teulings, and Rens, 2008) [55, 56] use a robust system GMM estimator on a dynamic panel model to address heteroskedasticity and endogeneity, but it is difficult to justify the over-instrument issue that makes the estimation biased. We apply the simple LSDV regression on the dynamic panel model, where inequality at year $t-5$ is hypothesized to be correlated with inequality at year $t$ and the 5-year average data of explanatory variables. Generally, the cross-sectional dependence issue in panel data makes the fixed-effects estimation inconsistent when annual data are used. However, we use 5-year average panel data for explanatory variables, and the time series of all countries barely share common years, which may dramatically reduce, if not entirely eliminate, the inconsistency issue and serial correlation in the error terms. Moral-Benito (2013) [4] simulates a set of estimators in a small time horizon and with a large observation size for various settings. The author finds that the fixed-effects (or the LSDV) estimator outperforms the first-difference GMM and system GMM estimators in terms of Nickell bias [57] and nonstationarity in the lagged and predetermined regressors. Therefore, we will apply the simple fixed-effects LSDV method to dynamic models.

To avoid multicollinearity caused by the inclusion of both openness and international trade (exports and imports), we will let either one of the two variables, not both, be included in our specifications. In particular, exports and imports, not openness, are used to estimate explained GDP, and openness but not trade is used to explain inequality. Both current and 5-year lagged educational attainment, employment and labor share may be included in explaining inequality if these variables show significant effects in the regressions or in making the specifications with trended and fixed time forms comparable.

We consider many explanatory variables used in the literature, consider both trended and fixed time forms for all variables, and consider both static and dynamic specifications, and we apply POLS to the estimation across countries and LSDV to the estimation within countries. Hence, there are eight regressions for each measurement of income distribution, in which the four POLS regressions are for between-country estimation that does not include country or time terms in the regressions, and the other four LSDV regressions are for within-country estimation that includes country and time terms in the regressions. We consider seven statistical measurements of income distribution: Gini, MES, MID, MIS MPS, B20, and T20.

We proceed with the estimation in five steps. First, we run the regression (3.3.1) for each variable $x_{it}$ in the set of original explanatory variables X = $\{x_{it}\}$ to obtain the new data; $nx_{it}$, $nX$ = $\{nx_{it}\}$ is the set of explanatory variables with the new data.

Second, we estimate the data of explained GDP, $\widehat{GDP}_{it}$, which is fitted GDP, and unexplained GDP, $\widehat{e}_{it}$, which is the residual, by running OLS regression on the following specification (5.1):

$$GDP_{it} = \widehat{\beta}_0 + nX\widehat{\beta} + \widehat{\beta}_c i.country + \widehat{\beta}_y t(year) + \widehat{e}_{it}, \qquad (5.1)$$

$\widehat{GDP}_{it}$ and $\widehat{e}_{it}$ are the fitted values and residuals of the above regression, respectively. $nX$ includes nEmp, nG, nI, nK, nHR, nXT, nMT, nOpen, nPG, nFDI, ntey, nsey, npry, ncl, npi, and nlabsh and inequality measures (nGini, nMID, nT20, nMIS and nMPS) in the current and

last periods. $t(year)$ takes the form of either the dummy i. year or the level year, and the same is true for Eqs (5.2) ~ (5.4).

Third, we run OLS on the following regression (5.2) for $\widehat{GDP}_{it}$ to obtain its new data $n\widehat{GDP}_{it}$, the residual of the regression, which has been netted out of the country and year effects.

$$\widehat{GDP}_{it} = \widehat{\alpha}_0 + \widehat{\alpha}_1 i.country + \widehat{\alpha}_2 t(year) + n\widehat{GDP}_{it} \tag{5.2}$$

Then, $\widehat{e}_{it}$ are orthogonal to $\widehat{GDP}_{it}$, $n\widehat{GDP}_{it}$ and all other regressors $nx_{ijt}$; the bilateral and multicollinearity among $n\widehat{GDP}^2_{it}$, $n\widehat{GDP}_{it}$, $\widehat{e}_{it}$ and $nx_{ijt}$ will be dramatically reduced and even close to zero because the variables with new data have zero mean. We run LSDV regressions on the following two specifications—static and dynamic specifications and trended and fixed time variables—to estimate within-country effects:

$$Ineq_{it} = \alpha_0 + \gamma_1 f(\widehat{e}_{it}, n\widehat{GDP}_{it}) + nX\alpha + \alpha_c i.country + \alpha_y t(year) + u_i + \varepsilon_{it}, \tag{5.3}$$

$$Ineq_{it} = \gamma_0 nIneq_{it-1} + \alpha_0 + \gamma_1 f(\widehat{e}_{it}, n\widehat{GDP}_{it}) + nX\alpha + \alpha_c i.country + \alpha_y t(year) + u_i + \varepsilon_{it}, \tag{5.4}$$

$$E(u_i nx_{ijt}) = 0, E(\varepsilon_{it}\varepsilon_{jt})_{i \neq j} = 0, E(u_i\varepsilon_{jt}) = 0, \ \varepsilon_{it} \sim (0, \ \sigma^2) \tag{5.5}$$

Note that the dependent variable $Ineq_{it}$ uses the original data (e.g., Gini rather than nGini) because country and time terms are included in the equations, and capital stock, investment, government spending and openness in $nX$ take a ratio definition, which differs from (5.1). $f(\widehat{e}_{it}, n\widehat{GDP}_{it})$ is the quadratic function of $\widehat{e}_{it}$ and $n\widehat{GDP}_{it}$, and $nX$ includes all new variables $nx_{ijt}$ and the interactive terms, which are defined as follows:

$$OpsGDP = nOpnsh * n\widehat{GDP}, clGDP = nCL * n\widehat{GDP}, piGDP = npi * n\widehat{GDP}.$$

Finally, we run POLS on the following two specifications—static and dynamic specifications and trended and fixed time variables—to estimate between-country effects:

$$nIneq_{it} = \alpha_0 + \gamma_1 f(\widehat{e}_{it}, n\widehat{GDP}_{it}) + nX\alpha + \varepsilon_{it}, \tag{5.6}$$

$$nIneq_{it} = \gamma_0 nIneq_{it-1} + \alpha_0 + \gamma_1 f(\widehat{e}_{it}, n\widehat{GDP}_{it}) + nX\alpha + \varepsilon_{it},$$
$$E(\varepsilon_{it}\varepsilon_{jt})_{i \neq j} = 0, \varepsilon_{it} \sim (0, \sigma^2) \tag{5.7}$$

Note that the dependent variable $nIneq_{it}$ uses the quasi-cross-sectional data (e.g., nGini rather than Gini), and country and time terms are not included in the Eqs (5.6) and (5.7).

Simultaneity between inequality and the explained and unexplained GDPs is not an issue to concern in the regressions (5.6) and (5.7) because the main channels (country and year effects) of correlation among the variables have been removed in one hand, and the unexplained GDP is orthogonal to the explained GDP in the other hand; meanwhile, there could be some correlation between GDP and income distribution, hence it is reasonable to include measures of income inequality to explain GDP, or let them go into the unexplained GDP. Furthermore, if we drop all the measures of income distribution from the specifications of explaining GDP, then in the regressions the R-squared becomes much smaller than that including the income terms. The bilateral correlation coefficients between our income distribution measures and the explained (or unexplained) GDP are very small, which is want we expect to observe and might also demonstrate not much of the simultaneity.

**Table 12. Regression results of Gini by LSDV.**

| Variable | (1) | (2) | (3) | (4) | (5) | (6) | (7) | (8) |
|---|---|---|---|---|---|---|---|---|
| nEmp | -0.139* | -0.145* | -0.351*** | -0.379*** | -0.192** | -0.126 | -0.319*** | -0.306*** |
| **nPG** | -1.368** | -2.093*** | -1.461** | -1.452** | -1.145* | -1.615** | -1.485** | -1.342** |
| nHR | -0.225*** | -0.059 | -0.260*** | -0.139** | -0.211*** | -0.1 | -0.248*** | -0.107* |
| **OpsGDP** | -0.315** | -0.387** | -0.474*** | -0.549*** | -0.342* | -0.425** | -0.475*** | -0.491*** |
| L5.npry | -0.014* | -0.016* | 0.002 | -0.021** | -0.016* | -0.019** | -0.006 | -0.024*** |
| ntey | 0.027* | 0.027* | 0.015 | 0.044*** | 0.016 | 0.016 | 0.032* | 0.035** |
| npi | 0.015 | 0.002* | 0.030* | 0.002** | 0.024 | -0.01 | 0.016 | 0.002* |
| nOpnsh | 0.037*** | 0.015 | 0.055*** | 0.020* | 0.019 | 0.01 | 0.048*** | 0 |
| L5.nlabsh | 0.195*** | 0.127* | 0.092 | 0.192*** | 0.176** | 0.115 | 0.122* | 0.106* |
| nGsh | 0.124** | 0.229*** | 0.120* | 0.002 | 0.117* | 0.226*** | 0.126* | 0.052 |
| **nIsh** | 0.250*** | 0.151** | 0.220*** | 0.191*** | 0.257*** | 0.149* | 0.175** | 0.180*** |
| e | 0.059* | 0.005 | 0.149*** | 0.083** | 0.100** | 0.028 | 0.117*** | 0.047 |
| nGDP | -0.001 | 0.022 | 0.024 | -0.049** | -0.019 | -0.003 | -0.006 | -0.055*** |
| piGDP | 0.025 | 0.022 | 0.068* | 0.025 | 0.008 | 0.009 | 0.057 | 0.023 |
| clGDP | -0.080* | -0.04 | -0.036 | -0.090** | -0.026 | -0.015 | -0.038 | -0.053 |
| $e^2$ | | | | | -1.301* | -0.597 | -0.89 | 0.005 |
| $nGDP^2$ | | | | | -0.429** | -0.423** | -0.308* | -0.399** |
| Time | FE | FE | Trended | Trended | FE | FE | Trended | Trended |
| SorD | Static | Dynamic | Static | Dynamic | Static | Dynamic | Static | Dynamic |
| N | 171 | 170 | 171 | 171 | 169 | 170 | 171 | 172 |
| $R^2$_a | 0.991 | 0.99 | 0.989 | 0.992 | 0.988 | 0.989 | 0.987 | 0.991 |

Note: The dependent variable is the Gini coefficient with the original data. All explanatory variables for which the estimates are significant in at most two of the four regressions of each specification are not listed in the table; these variables are nCL, nKsh, L5.nGini, nFDI2, nFrdm, and nlabsh for all columns and L5.nGini for the dynamic specifications. Country dummies and time dummies (or trend) are included in all regressions. SorD denotes dynamic or static specification, and the same definition applies to later tables.

## 5.1 Robust determinants of income distribution within countries

We run regressions (5.3)~(5.4) for seven dependent variables: Gini, MES, MID, MIS MPS, B20, and T20. Table 12 below summarizes the LSDV regression results for Gini with two specifications: one includes $e^2$ and $nGDP^2$, and the other does not. The regression results of the other 6 statistics are presented in S4~S6 Tables in S1 Appendix.

We have three blocks of variables in the table. The lower block includes all GDP terms, none of which, except $nGDP^2$, has significant estimates in at least three of the four regressions of each specification. The middle block is for the variables that have significantly positive estimates in at least 3 of the 4 regressions from column (1) to column (4). The upper block is for the variables that have significantly negative estimates in at least three of the four regressions for each specification.

The estimates of $nGDP^2$ are significantly negative from columns (5) to (8), but the estimates for GDP are insignificant from columns (5) to (7), and the negative turning point of GDP in column (8) is out of the data sample! Furthermore, the regressions excluding $nGDP^2$ and $e^2$ show large differences; in the middle block, ntey, npi, and nOpnsh have significant effects in three of the four regressions from columns (1) to (4), but none of them do from columns (5) to (8).

Therefore, we do not find evidence for the Kuznets hypothesis within countries in the panel data. We will consider only the specifications from columns (1) to (4), which do not include

the quadratic terms $nGDP^2$ or $e^2$; the same rule applies to the regressions of the other 6 statistical measures.

We observe from columns (1) to (4) in Table 12 that in at least three of the four regressions, all explanatory variables in the middle block show significantly positive effects, and the variables in the upper block show significantly positive effects on the Gini coefficient, while the significant estimates of each variable are consistent in sign.

Note that using the same estimator, the regression results are robust for each dependent variable if any of the variables are dropped; these variables are not listed here for conciseness. It might be too strong an assumption to consider estimates robust only when they are significant in all four regressions (fixed or trended time, static or dynamic); hence, we propose the following definition.

**Definition 5.1** A variable is called a robust determinant of income distribution if it shows significant and consistent estimates in at least three of the four regressions (static and dynamic specifications, fixed and trended time) by LSDV or POLS.

We choose the most robust of the four regressions for each dependent variable if most of the robustly significant and consistent estimates are shared by the other regressions. Therefore, we choose columns (1) and (4); both have 10 robustly significant estimates in the middle and lower blocks. Considering the significant estimates in the lower block, regression (4) outperforms (1).

We choose the most robust estimator for other dependent variables in the same way, and we may end up with 2 or 3 estimators having the same number of robust estimates for each dependent variable, which can be either static or dynamic specifications. The difficulty of choosing the most robust of the four regressions arises when one estimate is insignificant in one regression and significant in the other three regressions. The related variables are nEmp, npi, nGsh and nOpnsh*nGDP (for details, refer to S4~S6 Tables in S1 Appendix).

There are three possible choices for a unique robust estimator for each dependent variable: the first is static, the second is dynamic, and the last has exactly the same number of robust estimates in two of the four regressions. However, there could be one insignificant estimate among nEmp, npi, nGsh and nOpnsh*nGDP, which may be significant in the other three regressions.

Table 13 below summarizes the robust determinants, in static specification, of the seven statistics of income distribution, where the estimates of npi for Gini and nEmp for MID are not significant but they are significant in the other three regressions. The nonrobust estimates are not listed to make the table concise (details refer to column (1) in Table 12 for Gini and S4~S6 Tables in S1 Appendix for other statistics).

Summarizing Table 13, we have obtained the following evidence for the robust determinants of income distribution within countries:

- Unexplained GDP has robustly negative effects on the bottom 20% and median income shares, a positive effect on the top 20% income share, and a positive (slightly significant) effect on the Gini coefficient, which implies that relatively low income people could not deal well with economic uncertainty, but top-income people could.

- The overall effects of openness on inequality are robustly positive in underdeveloped countries but ambiguous in developed countries, while it has negative effects on the mean income share and mean population share and (weakly) positive effects on the middle 60% income share.

- Labor income share has robustly and consistently positive effects on income inequality; which is contradictory to the literature but presumably convincing because the estimates on

**Table 13. Robust determinants of income distribution within countries.**

| Variable | Gini | MIS | B20 | MES | T20 | MID | MPS |
|---|---|---|---|---|---|---|---|
| nEmp | -0.139* | | | | | 0.082 | |
| L5.nEmp | | **0.148***** | **0.090***** | 0.217*** | | | |
| **nGDP** | | **0.068***** | **0.028***** | **0.061***** | -0.084*** | | |
| clGDP | | | 0.032*** | 0.048*** | | | |
| nHR | **-0.225***** | | | 0.076*** | -0.170** | **0.118**** | |
| nPG | **-1.368**** | | | 0.780*** | | **1.490***** | |
| OpsGDP | -0.315** | -0.251** | | | | 0.203* | -0.452*** |
| nKsh | | | | | | | -0.017*** |
| L5.nsey | | | | 0.009*** | | | -0.009* |
| npry | | | | **0.040***** | | | |
| L5.npry | -0.014* | | | -0.013*** | | **0.025***** | -0.018** |
| nGsh | 0.124** | | -0.026** | | | | -0.089* |
| e | 0.059* | | -0.044*** | -0.063*** | 0.093** | | |
| **nIsh** | **0.250***** | -0.280*** | -0.112*** | -0.236*** | 0.242*** | | |
| L5.nlabsh | 0.195*** | | -0.056*** | -0.132*** | **0.233***** | | **0.164**** |
| ntey | 0.027* | | | | **0.045**** | -0.043*** | **0.039***** |
| L5.ntey | | | -0.015*** | | | | |
| nFDI | | | -0.011* | | 0.074** | -0.048** | |
| nFDI² | | **1.270***** | | | | -0.604* | |
| nCL | | | | -0.004*** | | -0.005* | |
| nFrdm | | | | -0.005* | | | |
| nsey | | | | -0.010*** | | | |
| nOpnsh | **0.037***** | | | | | | |
| npi | 0.015 | | | | | | |
| _cons | 0.471*** | 0.326*** | 0.031*** | 0.178*** | 0.513*** | 0.455*** | 0.671*** |
| Time | FE | Trended | FE | FE | Trended | FE | Trended |
| SorD | Static | Static | Static | Static | Static | Static | Static |
| N | 171 | 172 | 173 | 171 | 172 | 171 | 167 |
| R²_a | 0.991 | 0.957 | 0.987 | 0.997 | 0.979 | 0.981 | 0.96 |

Note: All estimates in the table are robust in at least three of the four estimations and the other estimates are not listed in the table. Highlighted estimates are robust in the four regressions. Shaded values denote inconsistent estimates of Gsh on Gini, MPS and B20. Estimates of piGDP and nlabsh are not significant for all (or not more than 1) 7 inequality statistics and not listed in the table.

different statistics of income distribution are consistent. A possible explanation is that the variation in labor income share is dominated by that of top skilled workers' earnings, which are counted in the top 20% income share.

- Investment share to GDP shows robustly and consistently positive effect on income inequality, while capital stock share to GDP has a strongly negative effect on mean-income population share but not much effect on other statistics. Therefore, capital formation rather than capital stock (development) raises income inequality within countries.

- Educational attainment plays a complex role in income distribution. Primary education has robustly negative effects on income inequality. Secondary education has robustly negative effects on the mean population share and could have neutral impacts on the median income share. However, tertiary education has robustly positive effects on income inequality.

- The government spending share to GDP does not show consistent signs on MPS and B20 in the regressions. *Gsh* has a robustly positive effect on the Gini coefficient but a negative effect on both MPS and B20.

The signs on the interactive terms of openness with Gini, MIS and MPS are considered to be consistent because the coefficient size of MPS is larger than that of MIS so that the negative effect on Gini is dominated by the fall of MPS. Five-year lagged primary educational attainment shows robustly negative effects on the Gini coefficient, which is in line with the negative effect on mean population share and positive effect on the middle 60% income share, even though the effect on median income share is very significant and negative because the effect is economically the least (-0.013) among the other three (-0.014, 0.025, -0.018). Note that negative impacts on inequality here imply either negative effects on the Gini coefficient, T20, and MPS or positive effects on B20, MIS, MID, and MES. The opposite applies to the meaning of positive impacts on inequality.

## 5.2 Robust determinants of income distribution across countries

We run POLS regressions on the quasi-cross-sectional data by Eqs (5.5) and (5.6). Table 14 below summarizes the regression results of nGini with the specification without the term $nGDP^2$.

Similar to Table 12, we group all explanatory variables into 3 blocks, where the upper and middle blocks are for the variables having robust estimates in at least 3 of the 4 regressions, and the lower block includes the estimates of GDP terms and the variables that are significant in exactly 2 of the 4 regressions. Columns (1) and (2) have the same number of robust determinants and column (2) has a higher explanatory power since the specification is dynamic. However, column (1) has two more significant estimates in the lower block than column (2). We choose the most robust estimator by looking at the number of significant estimates first and the adjusted $R^2$ second; therefore, we choose column (1), the static and trended regression, as

**Table 14. Regression results of nGini by POLS.**

| Variable | (1) | (2) | (3) | (4) |
|---|---|---|---|---|
| **nEmp** | -0.245*** | -0.246*** | -0.196** | -0.175** |
| **L5.npry** | -0.016* | -0.014* | -0.022** | -0.018* |
| nOpnsh*nGDP | -0.319* | -0.365** | -0.252 | -0.315* |
| **nIsh** | 0.245*** | 0.241*** | 0.302*** | 0.258*** |
| L5.nlabsh | 0.217** | 0.136* | 0.157* | 0.11 |
| nlabsh | 0.209** | 0.147* | 0.123 | 0.121 |
| nPG | -0.743 | -0.846 | -1.451* | -1.202* |
| nKsh | -0.011** | -0.010** | 0 | -0.002 |
| nCL*nGDP | -0.034 | -0.041 | -0.065 | -0.067 |
| npi*nGDP | 0.046 | 0.049 | 0.054 | 0.047 |
| nGDP | -0.043* | -0.034 | -0.016 | -0.024 |
| e | 0.082* | 0.053 | 0.106** | 0.073 |
| Time | Trended | Trended | FE | FE |
| SorD | Static | Dynamic | Static | Dynamic |
| N | 173 | 173 | 173 | 173 |
| $R^2$_a | 0.313 | 0.431 | 0.277 | 0.408 |

Note: The dependent variable is nGini for all regressions. The other explanatory variables not listed are nCL, nFrdm, nFDI, $nFDI^2$, npi, nOpnsh, ntey, L5.nsey, and nGsh; the 5-year lagged dependent variables in columns (2) and (4) are not listed; country and time variables are not included in the POLS regressions.

**Table 15. Robust determinants of income distribution across countries.**

| Variable | nGini | nMIS | nB20 | nMES | nT20 | nMID | nMPS |
|---|---|---|---|---|---|---|---|
| clGDP | | | | 0.044* | | | |
| L5.nEmp | | 0.133* | **0.083***** | **0.184***** | -0.217*** | | |
| **nGDP** | | | **0.020***** | **0.067***** | **-0.086***** | **0.056***** | |
| nHR | | | 0.045** | **0.100**** | -0.109* | **0.103**** | |
| OpsGDP | -0.319* | | | | -0.234* | 0.162 | **-0.397***** |
| nEmp | **-0.245***** | | | | | **0.136**** | **-0.225***** |
| L5.npry | **-0.016*** | | | | | **0.018***** | **-0.015*** |
| nPG | | | | | | **0.832*** | |
| nKsh | | -0.007* | | | | | **-0.012***** |
| ntey | | | | -0.018 | | **-0.036**** | **0.027*** |
| nCL | | | -0.003** | **-0.006**** | 0.007* | -0.007** | |
| **nIsh** | **0.245***** | **-0.238***** | **-0.056***** | **-0.136***** | **0.159***** | **-0.067*** | |
| e | | -0.078* | **-0.043***** | **-0.075**** | 0.069* | | |
| L5.nlabsh | 0.217** | | | | **0.182***** | | |
| L5.ntey | | | -0.015** | | | | |
| nOpnsh | | | | -0.016* | | | |
| Time | Trended | Trended | FE | Trended | Trended | Trended | Trended |
| SorD | Static | Static | Static | Static | Static | Static | Static |
| N | 173 | 173 | 173 | 173 | 173 | 173 | 173 |
| R2_a | 0.313 | 0.252 | 0.315 | 0.424 | 0.455 | 0.439 | 0.259 |

Note: Non-robust estimates and non-robust variables are not listed in the table. Highlighted estimates are robust in the 4 regressions.

the most robust estimator for nGini. Similar to the LSDV regression results, the most robust estimator for a dependent variable may not be unique in terms of the number of robust estimates.

Table 15 below summarizes the static regression results of all 7 dependent variables by POLS. More detailed results of nGini refer to Table 14, and the detailed results of other statistics are presented in S7~S9 Tables in S1 Appendix.

Our regressions (see S7~S9 Tables in S1 Appendix) consistently show that labor income share is positively associated with income inequality across countries. Specifically, increasing labor income share is significantly associated with increasing top 20 income share and the Gini coefficient, meanwhile, the lower income shares (B20, MIS, MES, MID) are consistently (but insignificantly) decreasing; which is reasonable because labor income share accounts for the labor incomes of the top income earners, which dominate labor income share and the top 20 income share as well, while relatively speaking the lower labor incomes do not change much comparing to the changes of top labor incomes. This finding is contradictory to the related studies in the literature, all of which suffers the endogeneity issue of panel data and is often caused by the missing variables that are always highly correlated with some explanatory variables.

Comparing Tables 13 and 15, we can see that

- Some variables are robust determinants of income distribution for both within and across countries, among which investment and GDP are still the two factors impacting income distribution more robustly than other factors do, labor income share is still positively associated with income inequality, investment but not capital stock is significantly and positively associated with income inequality, etc.

- Some variables are robust determinants of a statistic within but not across countries, and vice versa.

- In particular, some variables (secondary education, financial development, freedom, government spending share to GDP, and inflation) do not show robustly significant effects on any income statistics across countries.

- Combine Tables 13 and 15 (see S10 Table in S1 Appendix) together and drop the estimates and/or variables that are not robust to the regressions in both of the two tables, then we are left with the robust determinants of income distribution across and within countries:

  1. The last period of employment, working hours, and explained GDP, interactive term of openness with GDP, and last period's primary education are the five most robust determinants of decreasing income inequality, each of which shows robustly significant effects on three of the seven measures of income distribution; the other four factors (current employment, population growth, capital stock, and civil liberty with GDP) show the effects of decreasing income inequality on one or two of the seven measures of income distribution.

  2. Investment, current tertiary education and unexplained GDP are the three most robust determinants of increasing income inequality, in which investment shows the effects on five of the seven measures and its effect economically outperforms all other factors, unexplained GDP and current tertiary education show the effects on three of the seven measures; the other three factors (last period's tertiary education, last period's labor income share and civil liberty) show the effects on one or two of the seven measures of income distribution.

  3. There are some findings that differ from the related literature. For instance, investment is economically and statistically the most robust determinant (The economic size of the estimate of investment is shown to be dominating in the simulation of subsection 5.3); last period's labor income share shows robustly significant and positive effects on the Gini coefficient and top 20 percent income share; primary and tertiary education, rather than secondary education, show robustly significant effects on income distribution.

### 5.3 Is it capitalism that has led to rising income inequality?

Piketty (2014, 2015) [58, 59] states that a higher rate of return on capital than growth rate could lead to a permanent increase in income inequality, which has been harshly criticized by mainstream economists (Acemoglu, 2015; Mankiw, 2015; Ray, 2015) [60–62]. We discuss this topic by simulating the total marginal effects of GDP, capital stock and investment on income distribution.

We do not take GDP growth and the rate of return on capital as explanatory variables in our regressions, but we have the GDP (linear and quadratic) terms, capital stock (share to output), investment (share to output), and the interactions of GDP with civil liberty, openness and inflation to account for the total effects of capitalism. GDP and its interactive terms explain growth effects, and capital stock and investment explain the effects of the rate of return on capital. We also consider unexplained GDP as one of the factors of capitalism since it is part of the total output.

Table 16 below summarizes the regression results of the GDP terms, capital stock and investment by LSDV and POLS using static specification for the Gini coefficient, the bottom and top 20% income shares and the mean income share, which are from the regressions in

**Table 16. Regression of income distribution measures on GDP and capital terms by LSDV and POLS.**

| Variable | Gini | MIS | MPS | B20 | nGini | nMIS | nMPS | nB20 |
|---|---|---|---|---|---|---|---|---|
| e | **0.059**\* | -0.073\*\*\* | -0.03 | **-0.043**\*\*\* | 0.082\* | **-0.086**\*\* | -0.025 | **-0.049**\*\*\* |
| GDP | -0.001 | **0.067**\*\*\* | -0.024 | **0.029**\*\*\* | -0.043\* | -0.007 | -0.037\* | **0.028**\*\*\* |
| clGDP | -0.080\* | 0.014 | -0.037 | **0.032**\*\*\* | -0.034 | -0.007 | -0.026 | 0.007 |
| OpsGDP | **-0.314**\*\* | **-0.251**\*\* | **-0.439**\*\*\* | 0.025 | **-0.319**\* | -0.189 | **-0.397**\*\*\* | 0.025 |
| piGDP | 0.025 | -0.001 | 0.019 | 0.006 | 0.046 | -0.003 | 0.008 | 0.002 |
| nKsh | 0.005 | -0.004 | **-0.018**\*\*\* | -0.001 | -0.011\*\* | **-0.013**\*\*\* | **-0.012**\*\*\* | 0.001 |
| nIsh | **0.250**\*\*\* | **-0.280**\*\*\* | 0.106\* | **-0.111**\*\*\* | **0.245**\*\*\* | **-0.199**\*\*\* | 0.058 | **-0.061**\*\*\* |
| Time | FE | Trended | Trended | FE | Trended | FE | Trended | Trended |
| N | 170 | 171 | 171 | 172 | 173 | 173 | 173 | 173 |
| $R^2\_a$ | 0.99 | 0.957 | 0.963 | 0.987 | 0.313 | 0.307 | 0.259 | 0.379 |

Note: All regressions are in static specification. Highlighted coefficients are robustly significant in at least 3 of the 4 regressions.

subsections 5.1 and 5.2. The regression results of the other three statistics (MID, T20 and MES) for the simulation are relegated to S11 Table in S1 Appendix.

The regression results in Table 16 show that investment share to GDP is the most robust and strongest factor of all explanatory variables, and capital stock alone does not show significant effects on inequality; in particular, the effects of openness on inequality are negative in developed countries and positive in developing countries. It is reasonable to assume that the growth of GDP (ΔGDP) can be much smaller than the growth of investment share (ΔIsh) to GDP. Therefore, investment strongly dominates all six other explanatory variables regarding their marginal effects on income distribution.

We perform experimental simulations using the results in Table 16 and take the total derivative of the dependent variable with respect to the GDP terms, capital and investment. Eq (5.3.1) below shows the total marginal effects of nGini by Table 16:

$$\Delta \text{nGini}_{POLS} \approx 0.082\Delta e - (0.043 + 0.034\text{nCL} + 0.319\text{nOpnsh} - 0.046\text{npi}) * \Delta \text{nGDP}$$
$$- 0.011\Delta \text{nKsh} + 0.245\Delta \text{nIsh} \tag{5.3.1}$$

While calculating the total marginal effects for each dependent variable, we consider multiple growth scenarios which are combinations of value assignment for the changes in GDP, capital and investment terms and the levels of the three explanatory variables (nCL, nOpnsh, npi). Table 17 below shows the statistics of these variables.

Putting the assigned values of the related variables into Eq (5.3.1) yields the two values of ΔGini and ΔnGini. We perform the same simulation for the other dependent variables using

**Table 17. Variable statistics for value assignment.**

| Time Form | Trended | | | Fixed Effect | | |
|---|---|---|---|---|---|---|
| Variable | Std. Dev. | Min | Max | Std. Dev. | Min | Max |
| e | 0.0399 | -0.1122 | 0.1274 | 0.0382 | -0.1205 | 0.1180 |
| nGDP | 0.0889 | -0.2978 | 0.3295 | 0.0873 | -0.2897 | 0.2962 |
| nKsh | 0.5298 | -1.6602 | 2.2822 | 0.4845 | -2.0216 | 1.8833 |
| nIsh | 0.0400 | -0.1348 | 0.1929 | 0.0381 | -0.1302 | 0.1782 |
| nCL | 0.4997 | -1.3642 | 2.7284 | 0.4871 | -1.3042 | 2.7772 |
| nOpnsh | 0.1813 | -0.6705 | 1.3368 | 0.1706 | -0.7020 | 1.3064 |
| npi | 1.4133 | -4.0780 | 25.1326 | 1.3905 | -3.9683 | 24.3869 |

**Table 18. Simulation results.**

|  | ΔGini<0 | ΔMES>0 | ΔB20>0 | ΔMIS>0 | ΔMPS<0 | ΔMID>0 | ΔT20<0 |
|---|---|---|---|---|---|---|---|
| I | 0.005 | 0.206 | 0.222 | 0.097 | 0.894 | 0.339 | 0.325 |
| II | 0.050 | 0.215 | 0.336 | 0.128 | 0.814 | 0.364 | 0.327 |
|  | ΔnGini<0 | ΔnMED>0 | ΔnB20>0 | ΔnMIS>0 | ΔnMPS<0 | ΔnMID>0 | ΔnT20<0 |
| I | **0.403** | **0.465** | 0.287 | 0.000 | 0.964 | **0.856** | **0.595** |
| II | **0.449** | **0.440** | 0.288 | 0.001 | 0.865 | **0.789** | **0.545** |

Note: The value in a cell denotes the probability of the marginal effects of growth on income statistics being positive (negative). There are two growth scenarios: Scenario I is an increase of 0.1 times of standard deviation in GDP, I and K, and a level of one time the standard deviation in nCL, nOpnsh and npi; scenario II is an increase of two times the standard deviation in GDP, I and K, and a level of three times standard deviation in nCL, nOpnsh, and npi.

LSDV and POLS estimators. We run the simulation experiments one million times and count the times that the total marginal effects are positive or negative to obtain the probability of the event. For each statistic, the simulation results are very similar for different growth scenarios, hence, we report only the results of two growth scenarios, one is that the changes in capital variables (e, GDP, K, I) take 0.1 times standard deviation of their data and the levels of civil liberty, openness and inflation take a value in an interval centered at zero and has a length of one standard deviation of their data; the other is for 2 and 3 standard deviations of the two sets of variables, respectively. The second scenario covers almost all of the observations of the dataset. We also consider the negative growth cases of capital terms, which generate exactly the opposite results to the positive growth cases, and the results are not reported here to be concise. Table 18 below summarizes the experimental results.

Table 18 shows that the total marginal effects of GDP, capital stock and investment on income inequality are very likely to be positive; that is, the effects on all statistics are in line with increasing income inequality within and between countries except the effects on the middle 60% and top quintile income shares between countries; especially, the bottom quintile and mean income shares were negatively impacted and mean population share was positively affected from growth within and across countries. Note that for both between and within countries the total marginal effects of growth on both MIS and MPS are very likely to be negative, which is in line with the regression results in Shao and Krause [6].

The story behind this conclusion might be that investment is the main drive of growth, and its profits had been mostly collected by capital holders and executive officers because within countries the top 20% income share is very likely to increase in growth, while the bottom 20%, mean, median, and middle income shares had rarely increased; while between countries the estimates are different on the middle 60% and top 20% income shares, which might explain that the skilled labor premium due to migration between countries, and capital incomes face higher uncertainty than that within countries.

Therefore, the development of capitalism rarely favored most of the population in relative terms within a country, even though there are exceptions across countries, where the middle 60% income share was very likely to be increased.

## VI Concluding remarks

Empirical studies on income distribution with panel data have not discussed the issues caused by multicollinearity, imprecise estimates and endogeneity when relevant variables are omitted and correlated with covariates concerned. We deal with these issues by transferring panel data into quasi-cross-sectional data, which either completely removes or dramatically reduces

multicollinearity among all explanatory variables and allows our estimates to be unbiased and consistent. We consider both fixed and trended time forms for the data and both static and dynamic specifications; therefore, we have four specifications to explore the robust determinants of income distribution either within countries by LSDV or across countries by POLS.

Our findings update the related literature on the main determinants of income distribution, which include development level, openness, educational attainment and labor income share, among which investment ratio to GDP and employment are found to be the most robust determinants within and across countries. Our estimates on labor income share show robustly and consistently positive effects on income inequality, which challenges the related literature. A possible explanation is that top wage earnings dominate the variation of labor income share, while increasing top wage earnings imply increasing the top income share and labor income share as well so that both labor income share and income inequality increase. Using the estimates, our simulation shows that in relative terms, the total marginal effects of capitalism development did not favor most of the people within a country.

This study tries to reduce the data correlation among variables by changing the data structure, but the structural correlation among productive factors has not been addressed, which deserves future exploration.

## Supporting information

**S1 Appendix.**
(XLSX)

**S1 File. This is the word file to explain the files of supporting information.**
(DOCX)

**S2 File. This is the excel file for raw annual data.**
(XLSX)

**S3 File. This is the dta file for the 5-year averaged data of all variables.**
(DTA)

**S4 File. This is the do file for the regressions with the data treated by trended time effects.**
(DO)

**S5 File. This is the do file for the regressions with the data treated by fixed time effects.**
(DO)

**S6 File. This is the do file for replication.**
(DO)

**S7 File. This is the m-file for simulation.**
(M)

**S8 File.**
(M)

## Acknowledgments

I thank Professor Peter Edger and Fabio Canova for their helpful comments; I also thank the participants at the China Camphor 2020 Nan Chang meeting and Econometric Association 2020 Asia conference for their comments. Dr. Tsun-Feng Chiang helped to calculate the data of MPS and MIS.

## Author Contributions

**Conceptualization:** Liang Frank Shao.

**Data curation:** Liang Frank Shao.

**Formal analysis:** Liang Frank Shao.

**Funding acquisition:** Liang Frank Shao.

**Investigation:** Liang Frank Shao.

**Methodology:** Liang Frank Shao.

**Resources:** Liang Frank Shao.

**Software:** Liang Frank Shao.

**Supervision:** Liang Frank Shao.

**Validation:** Liang Frank Shao.

**Writing – original draft:** Liang Frank Shao.

**Writing – review & editing:** Liang Frank Shao.

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
