## [Decision Letter · Decision Letter 0]

1 Apr 2021

PONE-D-21-01207

Robust Determinants of Income Distribution across and within Countries

PLOS ONE

Dear Dr. SHAO,

Thank you for submitting your manuscript to PLOS ONE. After careful consideration, we feel that it has merit but does not fully meet PLOS ONE’s publication criteria as it currently stands. Therefore, we invite you to submit a revised version of the manuscript that addresses the points raised during the review process.

I have now received two referee reports on your submission. Both referees find some interesting points in your paper which contribute to the literature. They suggest a major revision and provide some specific comments.

I am happy to provide you with the opportunity to revise and then resubmit the paper. For such a re-submission I feel that you need to respond to the referees' concerns carefully. I should also state that it is my intention to refer the paper back to the referees after its resubmission.  

We look forward to receiving your revised manuscript.

Kind regards,

Dao-Zhi Zeng

Academic Editor

PLOS ONE

Journal Requirements:

Reviewers' comments:

Reviewer's Responses to Questions

**Comments to the Author**

1. Is the manuscript technically sound, and do the data support the conclusions?

Reviewer #1: Partly

Reviewer #2: No

2. Has the statistical analysis been performed appropriately and rigorously? 

Reviewer #1: No

Reviewer #2: No

3. Have the authors made all data underlying the findings in their manuscript fully available?

Reviewer #1: Yes

Reviewer #2: Yes

4. Is the manuscript presented in an intelligible fashion and written in standard English?

Reviewer #1: Yes

Reviewer #2: Yes

5. Review Comments to the Author

Reviewer #1: This paper empirical studies on income distribution by transferring panel data into quasi cross-sectional data. It is a topic of interest to the researchers in the related areas but there are still some problems which should be addressed:

（1）In general, compared with cross-section data, panel data has richer information and less multicollinearity.However, in this paper, in order to reduce the multicollinearity between the data, the panel data is converted into the form of cross-section data, which is very interesting. It is suggested that the author gives the rationality of this method and the advantages of this method compared with the existing methods in this paper, and gives the corresponding references.

（2）Research conclusions should be added to the abstract. And please condense abstract into one paragraph.

（3）In the introduction, the author elaborated on the research methods, research steps and research contributions, but ignored the introduction of income distribution. This paper should also explain the significance and importance of analyzing income distribution.

（4）The author should strengthen the explanation of the results. E.g: As mentioned in the paper, the correlation coefficient in Table 4 is less than that in Table 2, but the correlation (positive or negative correlation) among variables in Table 4 has changed compared with that in Table 2. It is necessary for the author to explain the reasons.

（5）Page 17, Line 357-358, "1.177 −358 1.104 + 0.001" the coefficient value does not correspond to Table 5. Please check it.

Reviewer #2: This paper tries to empirically investigate the factors that affect the income distribution across and within countries. The author first combines regressions with time dummies or country fixed effects to construct the “explained” and the “un-explained” GDP, and then regresses both variables on factors like investment or trade openness, which allows the author to find the “robust” determinants of income distribution.

I have the following questions related to the paper.

First, in terms of methodology, this paper is not the first one to predict “explained” or “un-explained” GDP from a regression that includes time or year fixed effects along with additional control variables. Hence it’s not appropriate that the paper made a contribution in the methodology of GDP or income distribution estimation.

Second, I find it quite confusing when the author adds the measure of income inequality in the first step of GDP estimation, and then regresses the measure of income inequality on the GDP measure obtained from step 1. In doing so, it is quite likely that the estimation procedure would face simultaneous causality issue.

Third, since the number of observations is around 500 and the WIID database covers more than 200 countries, a brief summary of country and year coverage would be helpful. In particular, is the result in this paper affected by the choice of countries under observation or not?

Fourth, some findings highlighted in this paper are somehow strange. For instance, the author argues that labor income share positively affects the level of income inequality. Observations from existing literature, however, indicate that one of the main reasons for the increasing income inequality across many countries is related to the decreasing (increasing) share of labor (capital) income.

Finally, some part of the paper, for instance, the literature review on factors affecting income distribution, or the comparison with other papers in Section 4, could be shortened.

6. PLOS authors have the option to publish the peer review history of their article (what does this mean?). If published, this will include your full peer review and any attached files.

Reviewer #1: No

Reviewer #2: No

---

## [Author Response · Author response to Decision Letter 0]

15 Apr 2021

Dear Reviewers, 

I am delighted to be given the chance to undertake revisions to my manuscript “Robust determinants of income distribution within and across countries”. 

I highly appreciate your comments and suggestions, to which I reply one by one in the following (responses highlighted in yellow and italic). As will become clear, I have revised the manuscript substantially in line with your comments. 

Best regards,

Liang Frank Shao

Reviewer #1: 

（1）In general, compared with cross-section data, panel data has richer information and less multicollinearity. However, in this paper, in order to reduce the multicollinearity between the data, the panel data is converted into the form of cross-section data, which is very interesting. It is suggested that the author gives the rationality of this method and the advantages of this method compared with the existing methods in this paper, and gives the corresponding references.

Ans. Thanks for this comments. To explicitly stress rationality and advantages, I add the following expression after the second paragraph:

“The rationality of this method is [1，2] that each variable with the panel data set has been partitioned into two orthogonal parts, one is for the variable itself, the other is for the time and section effects, the full information of the panel data can be still reserved when both parts are considered in a regression. The advantage is that we still consider all covariates, among which multicollinearity has been effectively reduced so that estimates become more robust and precise comparing the existing tools.”

 One reference for the methodology is the classic textbook [1] by Jeffrey M. Wooldridge (see page 356 and 359); Makram and Shao [2] used the similar way to measure redistribution (see equation (3) at page 258).

（2）Research conclusions should be added to the abstract. And please condense abstract into one paragraph.

Ans: Thanks for the suggestion and I have shortened it into one paragraph as follows and the conclusions are described as the contributions (Since it has been studied to partition GDP into explained and unexplained parts in the literature, I have deleted the expression about this part): 

“Multicollinearity widely exists in empirical studies, which leads to imprecise estimation and even endogeneity when omitted variables are correlated with any regressors. We apply an innovative strategy, different from the usual tools (instrumental variable, ridge regression, and least absolute shrinkage and selection operator), to estimate the robust determinants of income distribution. We transform panel data into (quasi-) cross-sectional data by removing country and time effects from the data so that all variables become zero mean and orthogonal to the country dummies and time variable, and multicollinearity becomes very low or even disappears with the quasi-cross sectional data in any specifications regardless of country dummies and time variable being included or not. Our contribution is threefold. First, we build a general method to address the multicollinearity issue in the empirical studies of income distribution, which is to isolate the common contents of correlated variables and ensures robust estimates in different specifications (dynamic or static specifications) and estimators (within- or between-effects estimators). Second, we find no evidence for the Kuznets hypothesis within and across countries; investment and employment are the most robust determinants of income inequality; meanwhile, labor income share shows robustly and consistently positive effects on income inequality, which challenges the related literature. Last, simulations with our estimates show that the total marginal effects of development (regarding GDP, capital stock and investment) on income inequality are very likely to be positive within and between countries except that the impacts on middle-60% and top-quintile income shares are not so likely within countries.”

（3）In the introduction, the author elaborated on the research methods, research steps and research contributions, but ignored the introduction of income distribution. This paper should also explain the significance and importance of analyzing income distribution.

Ans: Thanks for this suggestion and I have added the following paragraph in the introduction and the significance and important are mainly discussed in the subsections of literature review and replications:

“Panel data has been intensively and directly used in the empirical studies [1, 2, 3, 4] of income distribution. But multicollinearity has often been ignored in the literature so that omitted variable bias and endogeneity have been the main obstacles that are hardly well treated, and robust determinants of income distribution are still an open question in the literature.” 

（4）The author should strengthen the explanation of the results. E.g: As mentioned in the paper, the correlation coefficient in Table 4 is less than that in Table 2, but the correlation (positive or negative correlation) among variables in Table 4 has changed compared with that in Table 2. It is necessary for the author to explain the reasons.

Ans: Thanks for this suggestion and I have added the following highlighted part in the paragraph before Table 4:

“Table 4 below shows the bilateral correlations among the new variables corresponding to those in Table 2. The table shows that the bilateral correlations with the new (quasi-cross sectional) data have become much smaller than those with the original panel data in Table 2; for instance, the correlation coefficient between GDP and GDP2 is 1, but it is -0.007 between nGDP and nGDP2; this is because the data to be used in Table 4 have become the quasi-cross sectional data by removing the country and time effects from the original panel data to be used in Table 2, which removes the correlation effects of country and time in the original panel data and often changes the direction of the data as well. Note that what we care about here is the small size of the bilateral correlation coefficient in Table 4 so that the explanatory power (or say multicollinearity) is small too when we look at the regression of one variable on all other variables using the quasi-cross section data.”

（5）Page 17, Line 357-358, "1.177 −358 1.104 + 0.001" the coefficient value does not correspond to Table 5. Please check it.

Ans: Thanks very much for pointing out this typo. The numbers in Table 5 are the correct ones, and I have revised the expression as “1.25MPS-0.962MIS-0.000”.

Reviewer #2: This paper tries to empirically investigate the factors that affect the income distribution across and within countries. The author first combines regressions with time dummies or country fixed effects to construct the “explained” and the “un-explained” GDP, and then regresses both variables on factors like investment or trade openness, which allows the author to find the “robust” determinants of income distribution.

Ans：The explained and unexplained GDPs are used as explanatory variables to explain the robust determinates of income distribution. 

First, in terms of methodology, this paper is not the first one to predict “explained” or “un-explained” GDP from a regression that includes time or year fixed effects along with additional control variables. Hence it’s not appropriate that the paper made a contribution in the methodology of GDP or income distribution estimation.

Ans: Thanks for the comment and I have removed this expression in the abstract and conclusion part of the paper as following: 

“We apply twoan innovative strategyies, that isare different from the usual tools (instrumental variable, ridge regression, and least absolute shrinkage and selection operator), to estimate the robust determinants of income distribution. First, we extract the proportion of GDP unexplained by all known factors and denote it as unexplained GDP, which is orthogonal to the fitted (explained) GDP; both explained and unexplained GDP are used later to explain income distribution. Second, We transform panel data into (quasi-) cross-sectional data by removing country and time effects from the data for each variable, (including the explained GDP), which lets so that all variables become zero mean and orthogonal to the country dummies and time variable in the quasi-cross-sectional data.”

Second, I find it quite confusing when the author adds the measure of income inequality in the first step of GDP estimation, and then regresses the measure of income inequality on the GDP measure obtained from step 1. In doing so, it is quite likely that the estimation procedure would face simultaneous causality issue.

Ans: Thanks a lot for this comment. Indeed there could be simultaneity issue in the regression of income inequality on GDPs. I have added the following explanation after the specifications (5.6) and (5.7):

“Simultaneity between inequality and the explained and unexplained GDPs is not an issue to concern in the regressions (5.6) and (5.7) because the main channels (country and year effects) of the correlation among the variables have been removed in one hand, and the unexplained GDP is orthogonal to the explained GDP in the other hand; meanwhile, it is reasonable to include measures of income inequality to explain GDP, or let them enter the unexplained GDP. Furthermore, if we drop all the measures of income distribution from the specifications of explaining GDP, in the regressions the R-squared becomes much smaller than that including the measures (please see the table below). The bilateral correlation coefficients (please see the table below) between our income distribution measures and explained (or unexplained) GDP are very small, which is what we expect to observe and might also demonstrate no much of the simultaneity.” 

Bilateral correlation coefficient between GDP and income distribution measures

Correlation Detrended data De-fixed-effect data

Coefficient e GDP GDP2 e GDP GDP2

nGini 0.000 -0.035 -0.101 0.000 0.021 -0.087

nB20 -0.053 0.219 0.017 -0.040 0.205 -0.029

nT20 0.000 -0.198 -0.080 0.000 -0.165 -0.020

nMES -0.023 0.206 0.053 -0.022 0.184 -0.001

nMIS 0.000 0.023 -0.028 0.000 0.037 -0.022

nMPS 0.000 -0.163 -0.169 0.000 -0.107 -0.104

nMID 0.000 0.182 0.123 0.000 0.146 0.063

L5.nGini 0.000 0.057 -0.002 0.000 0.067 -0.007

L5.nB20 -0.106 0.074 0.067 -0.087 0.085 0.076

L5.nT20 0.000 -0.098 -0.025 0.000 -0.101 -0.012

L5.nMES -0.075 0.078 0.064 -0.068 0.075 0.052

L5.nMIS 0.000 0.016 0.101 0.000 0.036 0.133

L5.nMPS 0.000 -0.075 0.059 0.000 -0.049 0.125

L5.nMID 0.000 0.077 0.020 0.000 0.054 -0.024

N=180. The data including explained and unexplained GDPs are quasi-cross sectional.

Compare regression results for the explained GDP with and without measures of income distribution

Variable nGini nGini nMIS nMIS nT20 nT20 nMPS nMPS

e 0.082* 0.097** -0.078* -0.046 0.069* -0.006 -0.025 -0.037

GDP -0.043* -0.023 0.02 0.015 -0.086*** -0.072*** -0.037* -0.011

OpsGDP -0.319* -0.257* -0.136 -0.084 -0.234* -0.13 -0.397*** -0.248*

clGDP -0.034 -0.015 -0.043 -0.049* -0.035 -0.017 -0.026 -0.022

piGDP 0.046 -0.004 -0.039 0.008 0.025 -0.01 0.008 0.006

nEmp -0.245*** -0.184** -0.108 -0.142* -0.225*** -0.238***

L5.npry -0.016* -0.017** 0 0 -0.013 -0.009 -0.015* -0.010*

npry -0.002 -0.006 

nIsh 0.245*** 0.289*** -0.238*** -0.190*** 0.159*** 0.139** 0.058 0.006

L5.nlabsh 0.217** 0.129* 0.182*** 0.208*** 0.057 0.081

nKsh -0.011** -0.006 -0.007* -0.009** -0.006 -0.008* -0.012*** -0.014***

nlabsh 0.209** -0.015 -0.069 0.021 0.026 0.008

nHR -0.114* -0.035 0.066 0.003 -0.109* -0.063 -0.01 -0.029

nPG -0.743 -1.088* 0.709 0.658 -0.146 -0.004 -0.374 -0.092

L5.nsey -0.008 -0.012** -0.001 0.002 -0.005 -0.007 -0.006 -0.006

nsey -0.002 -0.001 

nGsh -0.003 0.053 -0.075 -0.028 0.05 0.065 -0.029 -0.045

ntey 0.024 0.022 0.030* 0.031* 0.027* 0.021

ncl 0.006 0.005 0.003 0.003 0.007* 0.009** 0.004 0.003

nfrdm 0.016* 0.016* 0.002 0.004 0.009 0.01 0.003 0.001

nOpnsh 0.025* 0.02 0.003 -0.009 0.01 0.019 0.002 0.002

nfdi 0.034 0.016 -0.014 -0.006 0.029 0.018 0.019 0.005

Fdisqr 0.138 0.246 0.302 0.015 0.27 0.029 0.185 0.039

npi 0.001 0 -0.001 0 0.001 0 0 0

L5.nEmp 0.133* 0.085 -0.217*** -0.152** 0.069 0.119*

L5.ntey -0.026 -0.02 0.027 0.026 

N 173 216 173 216 173 197 173 216

r2_a 0.313 0.157 0.252 0.161 0.455 0.403 0.259 0.208

Note: The explained GDP in the shaded column is specified by including measures of income distribution, and it is not in the other columns. The data is detrended, and the results of R2 are similar for the de-fixed-effect data.

Third, since the number of observations is around 500 and the WIID database covers more than 200 countries, a brief summary of country and year coverage would be helpful. In particular, is the result in this paper affected by the choice of countries under observation or not?

Ans: Thanks for the suggestion. I have added a table in the appendix to show the country and year information for the Gini coefficient, both the annual and 5-year average data are unbalanced. The table is as follows, which has been added by the end of the data summary subsection 3.2:

Table A2 Summary of the country and years of the 5-year averaged panel data

iso3 NGini year iso3 NGini year iso3 NGini year

MKD 1 2015 CZE 5 1995-2015 FRA 7 1985-2015

BLZ 2 1995-2000 DEU 5 1995-2015 IDN 7 1985-2015

CHN 2 1995-2005 ECU 5 1995-2015 JPN 7 1965-2010

GEO 2 2010-2015 EST 5 1995-2015 LUX 7 1985-2015

KOR 2 1995-2000 GRC 5 1995-2015 MYS 7 1980-2010

MUS 2 2010-2015 HND 5 1995-2015 NPL 7 1980-2010

NIC 2 2010-2015 KGZ 5 1995-2015 NZL 7 1985-2015

TUR 2 2010-2015 LVA 5 1995-2015 BGD 8 1975-2010

CHE 3 2005-2015 MDG 5 1995-2015 ESP 8 1975-2015

CYP 3 2005-2015 MNG 5 1995-2015 ISR 8 1980-2015

MLT 3 2005-2015 PER 5 1995-2015 NLD 8 1980-2015

MNE 3 2005-2015 PRT 5 1995-2015 PAN 8 1980-2015

SGP 3 2005-2015 PRY 5 1995-2015 CAN 9 1975-2015

SRB 3 2005-2015 ROM 5 1995-2015 HUN 9 1975-2015

ARG 4 2000-2015 RUS 5 1995-2015 NOR 9 1970-2015

CRI 4 2000-2015 SLV 5 1995-2015 FIN 10 1970-2015

DOM 4 2000-2015 SVN 5 1995-2015 ITA 10 1970-2015

GTM 4 2000-2015 VEN 5 1995-2015 SWE 10 1970-2015

HRV 4 2000-2015 CHL 6 1990-2015 USA 10 1970-2015

JAM 4 1990-2005 DNK 6 1990-2015 GBR 11 1965-2015

LTU 4 2000-2015 MDA 6 1990-2015 MEX 11 1965-2015

SVK 4 2000-2010 POL 6 1990-2015 PHL 11 1965-2015

SWZ 4 1995-2010 URY 6 1990-2015 THA 11 1965-2015

AUT 5 1995-2015 AUS 7 1985-2015 TWN 11 1965-2015

BGR 5 1995-2015 BEL 7 1985-2015 IRL 12 1975-2015

BLR 5 1995-2015 BRA 7 1985-2015 IND 13 1955-2015

COL 5 1995-2015 CSK 7 1960-1990 LKA 13 1955-2015

The number of countries observed the Gini index is 81, The toal observation of the Gini index is 486. NGini is the number of observations for each conntry. ISO3 is the country code.

The main idea of the paper is that quasi-cross sectional data gives more precise and robust results than panel data, which will not be affected by the choice of countries under observation. Furthermore, the results are from the data available so that its robustness requires the data to be random. Approximately speaking, selection bias is not a serious issue in the regressions since the data set covers most of the global countries and extended for decades.

Fourth, some findings highlighted in this paper are somehow strange. For instance, the author argues that labor income share positively affects the level of income inequality. Observations from existing literature, however, indicate that one of the main reasons for the increasing income inequality across many countries is related to the decreasing (increasing) share of labor (capital) income.

Ans:Yes, this finding is indeed contradictory to the literature, which has been stressed to be a novel contribution of the paper. I have added the following paragraph after Table 15 to explain this point: 

“Our regressions (see Table 12-15 and Table A151, A152 and A153) consistently show that labor income share is positively associated with income inequality across countries. Specifically, increasing labor income share is significantly associated with increasing top 20 income share and the Gini coefficient, meanwhile, the lower income shares (B20, MIS, MED, MID) are consistently (but insignificantly) decreasing; which is reasonable because labor income share accounts for the labor incomes of the top income earners, which dominate labor income share and the top 20 income share as well, while relatively speaking the lower labor incomes do not change much comparing to the changes of top labor incomes. This finding is contradictory to the related studies in the literature, all of which suffers the endogeneity issue of panel data and is often caused by the missing variables that are always highly correlated with some explanatory variables”

Actually, I replicate the fixed and random effects estimators using the panel data, the results are listed in the table below, which also shows exactly the negative effects of labor income share on inequality; obviously, the estimates of the replication is biased and inconsistent because all explanatory variables are correlated with the error term .

Variable Gini Gini T20 T20

labsh -0.096 -0.285*** -0.018 -0.210***

tey 0.062** 0.053* 0.064** 0.051**

HR 0.091 0.211** -0.125 0.073

Emp -0.105 -0.123 -0.318*** -0.188*

Ish 0.212** 0.066 0.213** 0.008

Gsh 0.177* 0.101 0.125 0.028

PG -1.242 0.255 -0.691 1.208*

FE RE FE RE FE

N 252 252 229 229

Note: The other covariates not listed are capital stock, primary education, secondary education, civil liberty, freedom index, openness, financial development index, and inflation, of which all estimates are insignificant.

Finally, some part of the paper, for instance, the literature review on factors affecting income distribution, or the comparison with other papers in Section 4, could be shortened.

Ans: Thanks for this suggestion and I have revised and shortened a lit bit of the related parts of the paper, and please refer to the revised version of the paper. The comparison with other papers in Section 4 could be largely shortened if the replication tables are dropped or relegated to the appendix, but I prefer to keep it to make the comparison clearer and easier to see the differences for audiences; especially it is friendlier to the audiences who are new to the topic.

---

## [Decision Letter · Decision Letter 1]

2 Jun 2021

Robust Determinants of Income Distribution across and within Countries

PONE-D-21-01207R1

Dear Dr. SHAO,

We’re pleased to inform you that your manuscript has been judged scientifically suitable for publication and will be formally accepted for publication once it meets all outstanding technical requirements.

Kind regards,

Dao-Zhi Zeng

Academic Editor

PLOS ONE

Additional Editor Comments (optional):

I have heard from both referees who had previously revised your manuscript and both are happy with the way you have addressed their major and minor concerns.

One referee only ask you to be more clear about how your results are different from the existing literature. I hope that you can do it in preparing the final version.

Reviewers' comments:

Reviewer's Responses to Questions

**Comments to the Author**

1. If the authors have adequately addressed your comments raised in a previous round of review and you feel that this manuscript is now acceptable for publication, you may indicate that here to bypass the “Comments to the Author” section, enter your conflict of interest statement in the “Confidential to Editor” section, and submit your "Accept" recommendation.

Reviewer #1: All comments have been addressed

Reviewer #2: All comments have been addressed

2. Is the manuscript technically sound, and do the data support the conclusions?

Reviewer #1: Partly

Reviewer #2: Partly

3. Has the statistical analysis been performed appropriately and rigorously? 

Reviewer #1: Yes

Reviewer #2: Yes

4. Have the authors made all data underlying the findings in their manuscript fully available?

Reviewer #1: Yes

Reviewer #2: Yes

5. Is the manuscript presented in an intelligible fashion and written in standard English?

Reviewer #1: Yes

Reviewer #2: Yes

6. Review Comments to the Author

Reviewer #1: (No Response)

Reviewer #2: The author has addressed most of the comments. I would suggest acceptance conditional on minor revisions in terms of the paper's clarifications. Most importantly, given that the results from this paper are somehow different from the existing findings, the author may carefully explain the differences and clarify the contribution of the paper.

7. PLOS authors have the option to publish the peer review history of their article (what does this mean?). If published, this will include your full peer review and any attached files.

Reviewer #1: No

Reviewer #2: No

---

## [Editor Report · Acceptance letter]

16 Jun 2021

PONE-D-21-01207R1 

Robust Determinants of Income Distribution across and within Countries 

Dear Dr. Shao:

I'm pleased to inform you that your manuscript has been deemed suitable for publication in PLOS ONE. Congratulations! Your manuscript is now with our production department. 

Kind regards, 

on behalf of

Dr. Dao-Zhi Zeng 

Academic Editor

PLOS ONE